# Sexual dimorphism in melanocyte stem cell behavior reveals combinational therapeutic strategies for cutaneous repigmentation

Luye An[1], Dahihm Kim [1], Leanne R. Donahue [1], Menansili Abraham Mejooli[2], Chi-Yong Eom[2], Nozomi Nishimura [2] & Andrew C. White [1] ✉

Vitiligo is an autoimmune skin disease caused by cutaneous melanocyte loss. Although phototherapy and T cell suppression therapy have been widely used to induce epidermal re-pigmentation, full pigmentation recovery is rarely achieved due to our poor understanding of the cellular and molecular mechanisms governing this process. Here, we identify unique melanocyte stem cell (McSC) epidermal migration rates between male and female mice, which is due to sexually dimorphic cutaneous inflammatory responses generated by ultra-violet B exposure. Using genetically engineered mouse models, and unbiased bulk and single-cell mRNA sequencing approaches, we determine that manipulating the inflammatory response through cyclooxygenase and its downstream prostaglandin product regulates McSC proliferation and epidermal migration in response to UVB exposure. Furthermore, we demonstrate that a combinational therapy that manipulates both macrophages and T cells (or innate and adaptive immunity) significantly promotes epidermal melanocyte re-population. With these findings, we propose a novel therapeutic strategy for repigmentation in patients with depigmentation conditions such as vitiligo.

Disorders of skin pigmentation exist across a broad spectrum. Vitiligo, one well-known autoimmune condition that has an incidence rate of 0.5–2% worldwide[1,2], is characterized by depigmented patches due to epidermal melanocyte loss. Vitiligo patients report a reduced quality of life, with a high incidence of psychosocial burden that can include anxiety, feelings of stigmatization, adjustment disorders, and relationship difficulties, among many others[3,4].

Various factors are involved in vitiligo pathogenesis including genetics, oxidative stress, innate and adaptive immunity, and environmental stimuli. These result in abnormal CD8 + T cell activity that ultimately causes epidermal melanocyte destruction[5]. Therefore, extensive studies have focused on targeting T cells as a potential therapy. Recently, several FDA-approved JAK inhibitors have been shown to improve repigmentation in vitiligo patients by inhibiting CD8+ T cell recruitment[6–9]. Even though phase II and III trials

demonstrate significantly improved skin pigmentation, response to these therapies varies among patients and full, durable repigmentation is rarely achieved[9].

To achieve durable repigmentation, treatment strategies that address both disease pathogenesis and promote repigmentation are likely to have greater efficacy[5]. Melanocyte stem cells (McSCs) residing in hair follicles represent a reservoir that can be utilized for repigmentation through phototherapy[10]. Numerous studies have revealed molecular mechanisms of McSC activation and differentiation under various conditions, including molecular mediators such as Mitf, the Tfap2 family, keratinocyte-derived Wnt signaling, Bmp signaling, and the Mc1R-cAMP pathway[11–17]. Besides keratinocyte-derived signals, recent studies brought more players into the picture. Macrophage-derived INF-γ[18], and sympathetic nerve-derived Adrb2[19] regulate follicular McSC activity as well. Additionally, inflammatory responses can

[1]Department of Biomedical Sciences, Cornell University, Ithaca, NY 14850, USA. [2]Meinig School of Biomedical Engineering, Cornell University, Ithaca, NY 14850, USA. ✉e-mail: acw93@cornell.edu

regulate McSC activity levels[20]. Together, these findings reveal a McSC regulatory network that involves numerous unique cell types and external stimuli. However, it is still challenging to translate these mechanisms into a therapy for vitiligo, as well as other non-autoimmune depigmentation conditions, due to limited drug options available to target pathways that can increase epidermal melanocyte repopulation.

Given these limitations, this study primarily focuses on identifying mechanisms of McSC recruitment to the epidermis and determining whether this process can be manipulated for the treatment of depigmentation conditions, such as vitiligo. To achieve this, we employed a precisely controlled murine UVB irradiation model based on phototherapy for vitiligo patients. Using this system, we identified essential cellular and molecular players that promote McSC proliferation and epidermal repopulation under UVB irradiation and determined that cyclooxygenase signaling or prostaglandin E2 supplementation significantly enhances this process. Furthermore, we examined the potential benefits of combining this approach with an FDA-approved immunosuppressant JAK inhibitor for vitiligo treatment. Taken together, these data identify an approach to repigmentation that may lead to better therapeutic outcomes for depigmentation conditions in patients.

## Results

### Tracking recruitment of hair follicle McSCs to the epidermis by UVB on non-pigmented mouse dorsal skin

Several excellent mouse models of vitiligo have been established to study T cell autoimmunity against epidermal melanocytes[21–23]. In addition to eliminating CD8 + T cell activity, replenishing the epidermal melanocyte population to regain pigmentation is another important aspect of vitiligo therapy. However, existing models of vitiligo are not ideal for studying the mechanisms of hair follicle McSC activation and migration for epidermal translocation. In one model, overexpression of Kitl (Stem cell factor) results in artificially high levels of melanocyte localization to the epidermis, thus potentially obscuring data to be collected for the natural UVB-induced process of translocation, as it would occur in human patients[21,22]. In another, auto-immunity results in induced follicle McSC loss and hair graying[23], such that the reservoir of McSCs is significantly depleted or no longer exists, thus eliminating the source of translocating melanocytes. In this study, we primarily focus on determining methods to promote epidermal melanocyte repopulation. Thus, we chose adult homeostasis of C57Bl/6 wildtype mice as a model to understand the mechanisms of UVB-induced stimulation of McSC activation and migration.

McSCs of the murine dorsal skin are located exclusively within the hair follicle stem cell niche (the bulge) and are not present in the IFE, similar to the distribution of skin melanocytes affected in human vitiligo patients. To visualize cutaneous McSCs, we utilized *Dct-rtTA; Tre-H2B-GFP* (*DG*) mice[18,24], in which McSC and differentiated melanocyte nuclei are labeled by H2B-GFP (Histone2B-Green Fluorescent Protein) following doxycycline administration[25] (Fig. 1A). To induce McSC migration to the IFE, we shaved the hair from the dorsal skin and irradiated with UVB (280–315 nm, dosage 2.2 J/m$^2$) three times at 8 weeks of age (Fig. 1A). This procedure results in a robust number of migrated melanocytes (Fig. 1B), but does not induce McSC DNA damage, or significant apoptosis 24 h (24 h) post irradiation (Supplemental Fig. 1A, B). These data are consistent with previous studies showing that short wavelength UVB exhibits low penetration through the epidermal layer[26], and indicate that McSC migration is likely not regulated by McSC DNA damage or related to wound healing[14,27].

The dynamics of McSC egress from the hair follicle niche and subsequent migration to the IFE following UVB irradiation are not well documented. To better understand this process and ultimately use this information to generate a signaling model that potentially includes multiple cutaneous cell types, we first interrogated UVB-induced McSC migration by fluorescence imaging. Following UVB exposure, we collected skin through a time course leading to maximal McSC appearance within the IFE and found that the first translocated McSCs could be identified on the day of the third UVB irradiation (day 5) (Fig. 1B). Following this, the number of translocated McSCs continued to increase until the endpoint on day 13.

Sectioned tissues only show snapshots of McSC migration amongst cells, but cannot directly demonstrate the migratory path of a single cell at differing timepoints[14]. To define the McSC migration path in vivo, we used *Tyr-CreER; LSL-tdTomato* (*TT*) mice to lineage trace McSCs by intravital imaging[28,29]. By taking advantage of two-photon microscopy, which provides high-depth penetrating visualization without invasive procedures, we mounted live *TT* mice under the microscope to monitor McSC migration from the same hair follicle once per day over three days, starting at day 6 post UVB exposure (Fig. 1C). Skin tissue contains a significant amount of collagen which generates a strong SHG (second harmonic generation) auto-fluorescence signal, shown with green pseudocolor. However, since cells are not present within the hair shaft tunnels, hair follicles can be detected by black holes in the images. Collagen autofluorescence combined with hair shaft tunnels was used to indicate the edges surrounding the hair follicles. All migrating melanocytes were found associated with hair follicles. Moreover, visualizing and monitoring a single McSC over one day showed distinct translocation from the middle of the hair follicle at day 7 to the epidermal surface at day 8 (Fig. 1D). In summary, McSCs receive signals that transmit UVB exposure information, and it requires several days for McSCs to begin and progress through proliferation and migration process along the epithelia of the bulge to the IFE.

At the collection timepoint, H2B-GFP+ melanocytes appear on the surface of the skin (Fig. 1E, F), whereas prior to irradiation, only autofluorescence from hair shafts is apparent. This observation suggested to us that leveraging the restricted imaging depth of an optical microscope to capture melanocytes that had translocated to the epidermis could facilitate better analysis and quantification across the entire UVB-irradiated area. To confirm the surface GFP+ signal was originating from translocated epidermal melanocytes and not hair bulge McSCs, we first examined longitudinal and transverse sections at day 13. Though hair bulge McSCs are visualized at a deep transverse section depth in both UVB and non-UVB skin, surface-depth transverse sections mirrored the pattern seen in whole mount, in which no GFP+ cells were evident in the non-UVB context (Fig. 1F). To further determine whether whole-mount visualization captures migrating McSCs, we utilized *Lgr6-CreER; lsl-tdTomato* mice. Lgr6 marks hair follicle cells above the hair bulge and sebaceous gland[30], a region that is often occupied by migrating McSCs following UVB-irradiation (Fig. 1B). tdTomato fluorescence on whole mount did not show the surface pattern identified through transverse section analysis and did not allow for focus on Lgr6+ cells that are found near the sebaceous gland (Supplementary Fig. 1C), confirming that only surface-level melanocytes are quantified by optical microscopy and demonstrating the utility of this method.

Additionally, to determine if epidermal McSCs are produced from an increased McSC pool generated through proliferation and migration from the bulge, we administered 5-Ethynyl-2'-deoxyuridine (EdU) to UVB-irradiated *DG* mice four times per day, beginning at day 3. Skin was collected from day 4 to day 8 and McSCs that were located between the bulge and the IFE were identified as migrating melanocytes. McSCs that had undergone proliferation were identified as GFP$^+$EdU$^+$ cells (Supplemental Fig. 1D). By sectioning the skin and quantifying the melanocytes associated with hair follicles, we found that within this time window, there was a consistent percentage of migrating melanocytes associated with each hair follicle (Supplementary Fig. 1E). Thus, we conclude that the melanocyte population found in the IFE following UVB exposure arises due to McSC proliferation and migration originating from the hair follicle bulge.

## Male mice exhibit increased McSC migration due to heightened skin inflammation

Human studies have found an association between lentigo maligna (a subtype of melanoma) and UV in men, but not in women[31], suggesting that melanocyte activities may differ between sexes. Additionally, there is a higher melanoma incidence rate in older men compared to older women[32,33]. Given this, we stratified our migration data based on sex. Surprisingly, we found that McSC migration efficiency was indeed

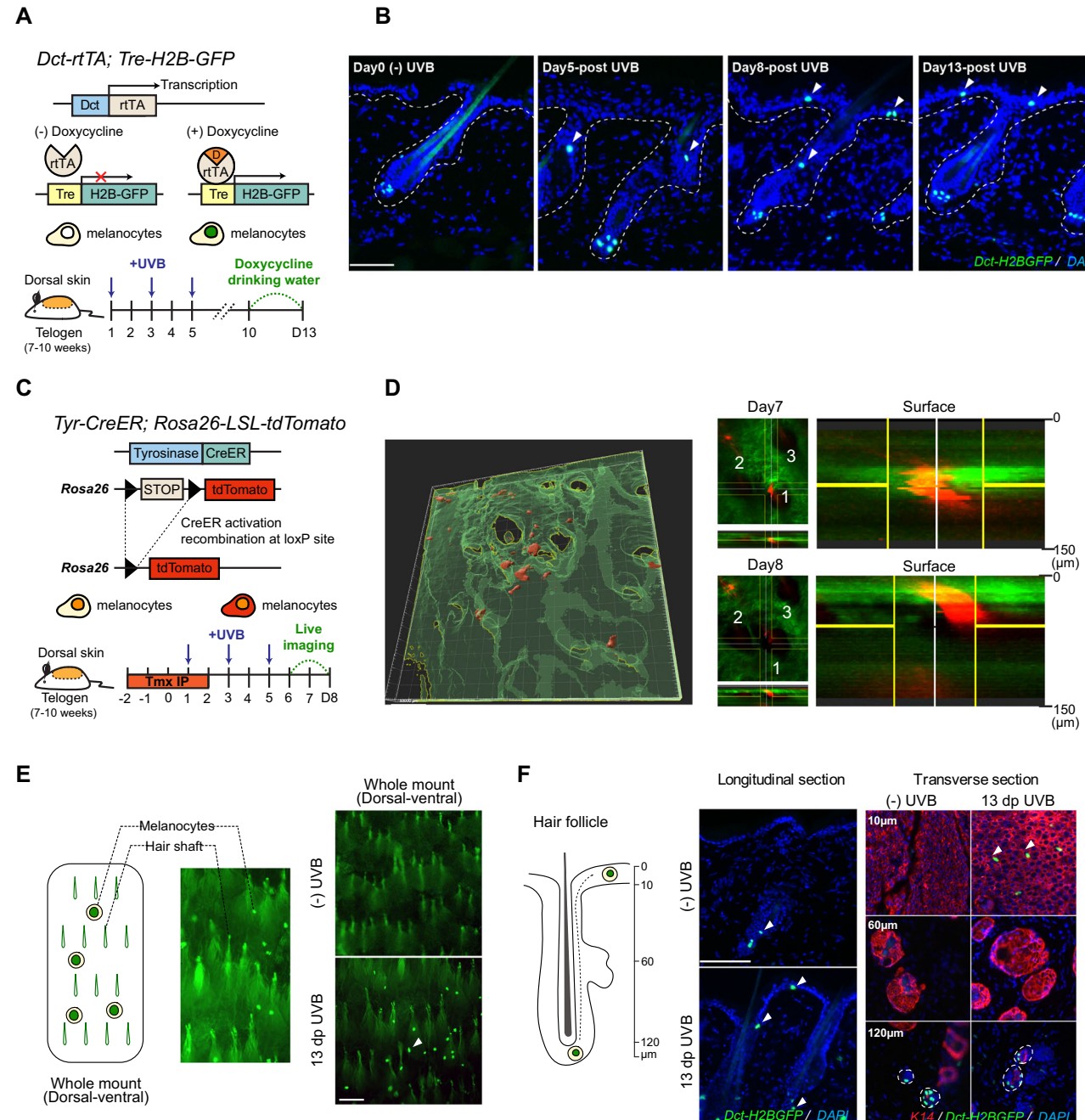

**Fig. 1 | Tracking recruitment of hair follicle McSCs to the epidermis by UVB on unpigmented mouse dorsal skin. A** Schematic diagram of *Dct-rtTA; Tre-H2B-GFP* (DG) mouse model (top), UVB irradiation timeline (bottom). **B** Representative images of longitudinal skin sections at different time points following UVB irradiation. Arrowheads denote McSCs, and dashed lines indicate the boundary between the epidermis and dermis. *n* = 3 males for each time point. **C** *Tyr-CreER; Rosa26-lsl-tdTomato* (TT) mouse model and imaging timeline. **D** Representative 3D rendered whole mount live skin image (left). Top-down (X–Y) perspective of raw images with the horizontal section view (X-Z) taken on the same area of skin at day 7 and day 8 (right). Red signal indicates tdTomato fluorescence (melanocytes); green signal indicates collagen SHG autofluorescence (skin tissue). 1–3 labels the locations of the same group of hair follicles. Enlarged images are the vertical view (Y-Z) of the area in the center of yellow cross. *n* = 3 males for each time point. **E** Schematic diagram of a dorsal-ventral view of whole mount dorsal skin and enlarged representative images (left). Representative images of whole mount dorsal skin images without UVB irradiation (top right) and with UVB irradiation (bottom right). As shown, without UVB irradiation, no H2B-GFP⁺ McSCs located in the hair bulge were detectable, and only epidermal H2B-GFP⁺ melanocytes were visible. **F** Schematic diagram of hair follicle and McSC migration (left), and representative images of non-UVB and UVB-irradiated skin with longitudinal sections (middle) and transverse sections at the skin surface (10 μm), middle (60 μm) and hair bulge (120 μm) depth. Keratin-14 (K14) labels keratinocytes in the interfollicular epidermis and hair follicles. Arrowheads indicate H2B-Gfp⁺ melanocytes. Dashed lines indicate the hair follicles. *n* = 3 males for no UVB and 13dp UVB group. Scale Bar: 100 μm. Mouse model diagrams were created with BioRender.com (**A, C**).

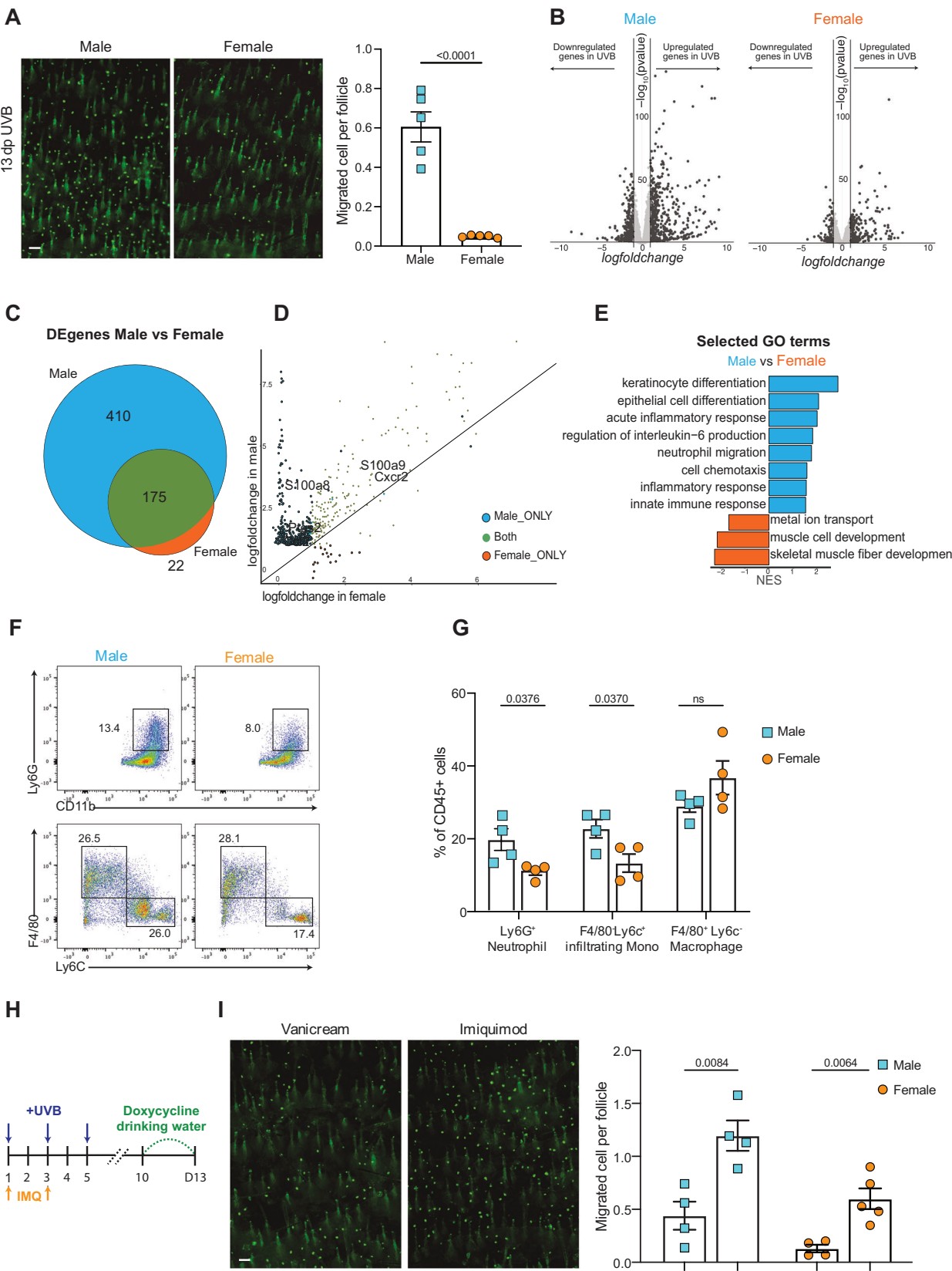

different between the genders of mice. Male mice have a strikingly higher migration rate than females (Fig. 2A).

To understand the unique responses to UVB irradiation between males and females, we collected whole dorsal skin from non-irradiated C57Bl/6N male and female controls and at 6 h following three UVB irradiations, a timepoint in which McSC proliferation and migration can be detected (Supplementary Fig. 1D). Tissues were processed and bulk mRNA sequencing was performed. The sequencing data showed that 585 upregulated genes were statistically significant in males and 197 upregulated genes were statistically significant in females (Fig. 2B,

**Fig. 2 | Male mice exhibit increased McSC migration due to heightened skin inflammation. A** Representative images of UVB-irradiated male and female skin 13 days following initial exposure (left) (*n* = 5 mice for males; *n* = 5 mice for females), and quantification (right). Quantification was performed by averaging the migrated melanocytes from at least 6000 hair follicles in each mouse. **B** Volcano plot of gene expression level changes between no-UVB and 6 h after third UVB (day 5) in male and female mice. Threshold: |log2FoldChanges| > 1, −log10 (pvalue) > 2. *n* = 4 male mice for UVB, *n* = 4 male mice for no-UVB, *n* = 4 female mice for UVB, *n* = 4 female mice for no-UVB. **C** Venn diagram showing the differentially upregulated genes following the threshold denoted in (**B**), in males and females. **D** Expression level of all differentially expressed genes in male and female samples. X and Y axes indicate the gene's lfc in females and males. **E** Selected Gene Ontology terms of GSEA analysis by comparing fold changes in relative gene expression in no-UVB control males and females. NES Normalized Enrichment Score. **F** Flow cytometry plots show CD11b⁺ Ly6G⁺ neutrophils, F4/80⁺ Ly6C⁻ macrophages, and F4/80⁻ Ly6C⁺ infiltrating monocytes from irradiated mouse skin collected on day 3. **G** Quantification of each cell type as a percentage of total CD45⁺ immune cells. *n* = 4 for each gender. **H** Experimental timeline for UVB irradiation under Imiquimod treatment. **I** Representative McSC migration images collected at 13 days following initial UVB exposure under control (Vani, Vanicream) and Imiquimod (IMQ) treatment (left), and quantification on both male and female mice (right). At least 6000 follicles per mouse were quantified. *n* = 4 for Vani-treated males, *n* = 4 for IMQ-treated males; *n* = 4 for Vani-treated females, *n* = 5 for IMQ-treated females. Scale bar: 100 µm. Statistics: significance calculated by student *t* test, two-tails with Welch's correction. Error bar: SEM (**A**, **G**, and **I**).

C, Supplementary Data 1). Furthermore, within the 197 upregulated differentially expressed genes (DEGs) in females, 175 of them were shared with male DEGs, but presented higher fold changes (Fig. 2D). Among these genes, we found many inflammatory response genes, including *S100a9*, *Ptgs2*, and *Ccl2*.

Cutaneous UVB irradiation results in both inflammation and immunosuppression[34,35]. However, immunosuppression induced by UVB is primarily achieved through antigen-presenting and T-cell function[36,37], whereas the pro-inflammatory response induced by UVB is largely associated with innate immune cell activity. To illustrate the different responses to UVB irradiation between males and females while minimizing the effect of transcriptome differences due to sex (Supplementary Fig. 2A), we calculated the logfoldchange (*lfc*) for each gene between males and females and performed gene set enrichment analysis. Gene Ontology analysis shows that male mice display profound keratinocyte differentiation, indicative of epidermal hyperproliferation, and a stronger inflammatory response (Fig. 2E, Supplementary Fig. 2B). To demonstrate at the cellular level that males present a stronger inflammatory response than females, we collected male and female UVB-irradiated skin at day 3, when neutrophils and macrophages are actively recruited to the skin (Supplementary Fig. 2C). Both neutrophils and monocyte infiltration are indicators of acute inflammation[38]. Thus, we digested the irradiated skin and performed flow cytometry to quantify the influx of these two populations (Fig. 2F, Supplementary Fig. 2D). We validated infiltration of neutrophils (CD45⁺, CD11b⁺, Ly6G⁺) and monocytes (CD45⁺, CD11b⁺, Ly6G⁻, Ly6C⁺) on skin collected at day 3. As expected, more neutrophils and monocytes were present in male skin compared to females (Fig. 2G). To determine if infiltrating innate immune cells regulate McSC migration, we performed the same irradiation procedure on NSG immunocompromised male and female mice, which show impaired myeloid cell recruitment compared to C57Bl/6N wildtype mice (Supplementary Fig. 2E). No difference in McSC migration was detected between sexes in NSG mice, demonstrating the importance of infiltrating immune cells in facilitating this phenotype. Interestingly, most NSG animals barely showed any McSC migration (Supplementary Fig. 2F).

Very interestingly, both post-inflammatory hyperpigmentation and hypopigmentation have been reported in human patients[39,40], indicating a role for inflammation in altering melanocyte and McSC activities. Our previous study showed decreased McSC migration following irradiation with concurrent dexamethasone treatment, an anti-inflammatory drug[20]. Thus, to determine if an increase in inflammation due to other means could enhance UVB-induced McSC activation and migration, we applied the pro-inflammatory agent imiquimod to irradiated dorsal skin. Imiquimod is a well-established topical agent that induces skin inflammation[41] by activating Toll-like receptor 7 to generate a psoriasis-like condition in mice[42]. To promote skin inflammation while not inducing the psoriasis phenotype, we applied 5% imiquimod cream on the UVB-irradiated area following the first and second UVB irradiation (Fig. 2H). Imiquimod with UVB significantly increased the migration rate compared to vehicle controls, while

imiquimod without UVB did not induce melanocyte epidermal translocation (Fig. 2I, Supplementary Fig. 2G). Taken together, our data indicate that male mice show a significantly increased inflammatory response compared to females and that increased inflammation by imiquimod can promote a higher number of migrated McSCs induced by UVB irradiation, while imiquimod-induced inflammation alone is not sufficient to elicit McSC migration. It is worth noting that there are many case reports showing vitiligo-like depigmentation as a side effect of imiquimod application[43–45]. We suspect that, in the context of our study, short-term inflammation promotes McSC epidermal repopulation while long-term imiquimod treatment on genital skin could cause McSC exhaustion, or cytotoxic T-cell infiltration[46]. Our findings suggest an alternative perspective for understanding the depigmentation side effect of imiquimod administration.

## Single-cell profiling reveals a transition to macrophages with a pro-inflammatory signature after UVB irradiation

To systematically define and dissect the murine cutaneous inflammatory response using our UVB-irradiation protocol and identify potential mediators of McSC migration, we performed single-cell mRNA sequencing on control non-irradiated dorsal skin and on dorsal skin five days following UVB-irradiation, the same timepoint at which whole skin was collected for bulk RNAseq. Given our focus on the inflammatory response, CD45+ magnetic beads were used to enrich immune cells and increase single-cell resolution. Within these profiles, we successfully identified five macrophage sub-populations, neutrophils, Langerhans cells, mast cells, two T cell sub-populations, and non-immune cells such as keratinocytes, fibroblasts, and neurons (Fig. 3A, Supplementary Fig. 3A). We then re-clustered immune cells in isolation and found that six subtypes of macrophages were distinguishable (Fig. 3B). Moreover, neutrophils, helper T cells, and early-stage T cells, but not cytotoxic T cells were enriched in UVB irradiated skin (Fig. 3C).

T cells have been identified as key players in epidermal melanocyte destruction in vitiligo patients and in a vitiligo mouse model[5]. However, following UVB irradiation, immunosuppression results from egress of the few T cells found in the dermis. Thus, T cells are unlikely to facilitate or promote McSC activation and migration in this context. In contrast, the UVB-mediated inflammatory response is generated by innate immune cells, including neutrophils and macrophages. Macrophages are the largest immune cell population found in inflamed skin tissue and they are key regulators of tissue homeostasis, damage repair, and regeneration (Fig. 3C)[47]. Given their abundance, diversity and long-lived presence in the skin prior to and after UVB-irradiation, we further investigated the functional heterogeneities between different macrophage subpopulations in non-UVB and UVB-irradiated tissue.

A recent study established a common framework for monocyte-derived macrophage activation, which included four macrophage activation paths in inflamed tissue: phagocytic, inflammatory, remodeling and oxidative stress[48]. Additionally, this published study used co-expression of *Retnla* and *Ear2* as unique markers of early-stage cells, and phagocytic macrophages were defined via high expression of

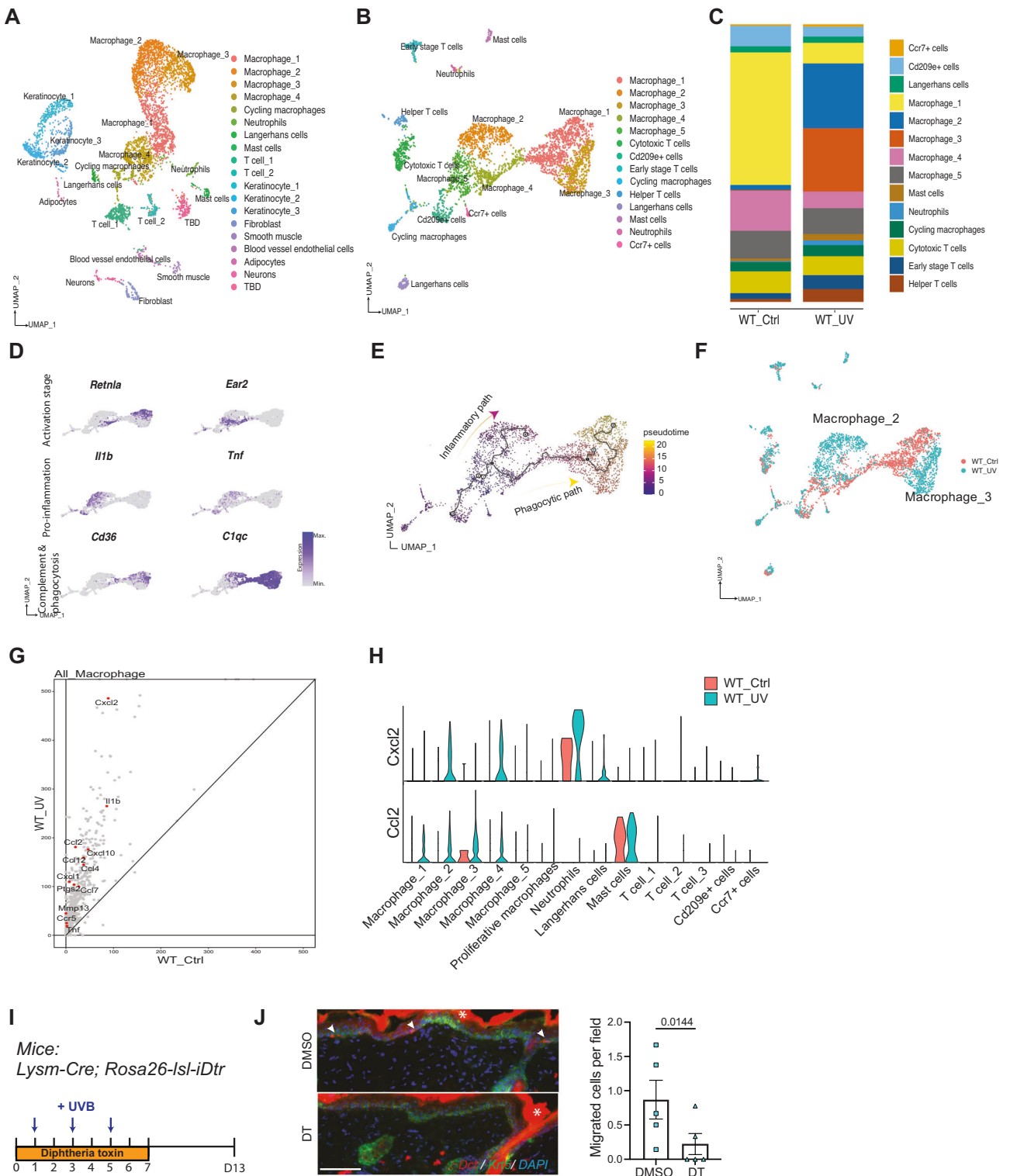

**Fig. 3 | Single-cell profiling reveals a transition to macrophages with a pro-inflammatory signature after UVB irradiation. A** UMAP of single cell clusters derived from WT male no-UVB (WT_Ctrl, $n = 1$) and WT male UVB day 5 (WT_UV, $n = 1$). **B** UMAP of all immune cells subclustered from cells shown in (**A**). **C** Cell composition of all immune cell types in WT_Ctrl and WT_UV. **D** Visualization of marker gene expression in macrophage subpopulations. **E** Pseudotime analysis of macrophage progression using Monocle 3. Starting timepoint is based on the expression of activation stage marker genes in (**D**). The inflammatory path and phagocytic path were determined based on the expression pattern of Pro-inflammation and Complement & Phagocytosis marker genes in (**D**). **F** UMAP of

immune cells based on sample. **G** Volcano plot of expressed genes in all macrophages. Pro-inflammatory cytokine genes are highlighted. **H** Violin plots showing Ccl2 and Cxcl2 mRNA expression levels in each immune subtype in WT control and WT UVB samples. **I** Timeline of diphtheria toxin (DT) administration and melanocyte migration assay using *Lysm-Cre; Rosa26-lsl-iDtr* mice. **J** Representative images of Dct staining on DMSO and DT-treated mice (left), and melanocyte migration quantification (right). $n = 5$ for DMSO and DT group. Arrowheads mark the migrated melanocytes. * marks autofluorescence from the stratum corneum. As the control group, *Lysm-Cre; Rosa26-lsl-iDtr* mice were treated with DMSO. Statistics: Welch's *t* test. Error bar: SEM (**J**). Scale bar: 100 μm.

macrophage alternative activation, complement cascade and antigen presentation markers (including *Mrc1, C1qc, H2-Aa*). We applied these markers and used Monocle 3[49] to analyze macrophages in our dataset based on activation phase (Fig. 3D, E). We identified two distinct macrophage activation pathways: an inflammatory path and a phagocytic path. Next, we investigated how these two cell states were distributed across samples. We determined that cluster Macrophage_2 and _3 are unique macrophage populations found only in the UVB irradiated sample (Fig. 3F), and that the Macrophage_2 population can be defined as the final stage of the inflammatory phase. Taken together, these data suggest that UVB irradiation alters the progression of macrophage activation, and that the irradiated skin microenvironment drives macrophage polarization towards the inflammatory path.

Furthermore, by comparing the bulk mRNA expression profiles between macrophages from UVB-irradiated and control skin, we found that macrophages from UVB skin expressed increased pro-inflammatory marker expression, including *Il1b, Tnf, Cxcl2, Ccl2*, and *Cxcl1* (Fig. 3G)[50]. To further investigate the cells expressing cytokines that recruit neutrophils and macrophages, we examined the expression level of Cxcl2 and Ccl2 in different immune cell types. Our analysis revealed increased Cxcl2 and Ccl2 expression in Macrophage_2-4 and neutrophils in the UV sample compared to the control sample (Fig. 3H). Additionally, utilizing CellChat analysis, we identified communication between various immune cell types through Ccl and Cxcl signaling networks (Supplementary Fig. 3B).

To determine if macrophages are an important component of the pro-inflammatory microenvironment that promotes McSC migration, we bred *Lysm-Cre* animals to the *Rosa26-lsl-iDtr* mice to specifically deplete monocytes and macrophages through diphtheria toxin (DT) administration (Fig. 3I). Following the same UVB irradiation procedure, we determined that DT-treated animals demonstrated less epidermal melanocyte migration compared with the DMSO-treated group (Fig. 3J). Overall, we found that macrophages promote UVB-induced McSC migration and that irradiation drives more macrophages toward an inflammatory phenotype.

## Enhanced prostaglandin signaling in males regulates McSC migration

To further identify the potential molecular mediators that drive the pro-inflammatory microenvironment following UVB irradiation, and also to obtain a sex-specific expression profile, we interrogated the transcriptomic distribution of pro-inflammatory genes and pathways between males and females. *Ptgs2* (the Cox-2 coding gene) and the downstream pathway member *Ptges* (PGE₂ synthase) were uniquely identified as significantly upregulated in males, suggesting an enhancement in prostaglandin signaling (Fig. 4A). Cox-2 and its downstream prostaglandins are well-known to be expressed and produced by macrophages under pro-inflammatory stimuli, such as LPS and IL-1, to mediate inflammatory responses[51]. Thus, we examined *Ptgs2* expression in our single-cell dataset. As expected, *Ptgs2* was expressed in pro-inflammatory Macrophage_2, as well as neutrophils, but not in phagocytic macrophages, indicating a mediator role for Cox-2 in promoting inflammation in this context (Fig. 4B).

Furthermore, from our bulk RNAseq data on UVB-irradiated male skin, we identified a higher level of upregulation in *Ptgs1, Ptgs2, Ptges, Ptges3*, and the arachidonic acid metabolism signaling pathway, all of which are necessary for production of prostaglandins molecules (Fig. 4C, Supplementary Fig. 4A). *Ptgs2*/Cox-2 has been shown to be expressed in epidermal keratinocytes as well after UVB irradiation, but not in non-irradiated skin[52,53]. Conversely, *Ptgs1*/Cox-1 is thought to be ubiquitously expressed at low steady-state levels in most cell types, and is considered generally unresponsive to external stimuli such as UVB[53].

To determine if macrophages express Cox-2, as well as other genes in the PGE₂ signaling pathway, immunofluorescence staining on UVB-irradiated male skin was performed to screen for colocalization

with PGE₂ synthases and receptors. We found that Ptges2, Ptges3, and Ptger2 colocalize with F4/80 macrophages, suggesting that macrophages can act as both a source and target of PGE₂ signaling and may play a key role in regulating McSC activity (Fig. 4D).

To determine if the increased Cox-2 activity in males contributed to higher UVB-induced migration, we deleted *Ptgs2*/Cox-2 globally using *Rosa26-CreER, Ptgs2^{f/f}; Dct-rtTA, Tre-H2B-GFP*[54–56] mice. As expected, we did see a significant decrease in McSC migration in global knockout male mice (Fig. 4E), which is consistent with our previous study showing the administration of dexamethasone decreases UVB-induced McSC migration[20]. These data suggested that increasing Cox-2 activity might increase McSC epidermal migration, which could be useful for clinical applications.

## Increased cyclooxygenase expression escalates McSC proliferation and epidermal repopulation

To determine if increased Cyclooxygenase activity could be used to promote McSC activation and migration, we bred *Rosa26-rtTA, Tre-Ptgs2* (*RP*) mice[57]. In this model, rat *Ptgs2* is overexpressed globally when mice are provided with doxycycline (Fig. 5A). To confirm that this allelic combination worked as expected, anti-Cox-2 staining demonstrated overexpression in both epidermal cells and immune cells within the dermis (Fig. 5B, Supplementary Fig. 5A). To confirm that Cox-2 enzymatic activity is also increased in this model system, we isolated UVB-irradiated skin from control and *RP* mice for ELISA analysis to quantify cutaneous prostaglandins levels. As expected for increased *Ptgs2*/Cox-2 expression, both PGE₂ and PGD₂ were significantly upregulated in *RP* mice (Fig. 5C, Supplementary Fig. 5B). To determine how Cox-2 overexpression affects inflammatory responses, we collected skin from *RP* mice on day 5 and examined immune cell infiltration by immunofluorescence staining. *RP* mice showed increased neutrophil infiltration compared to control mice, as indicated by anti-Ly-6G and anti-S100a9 staining (Fig. 5D, Supplementary Fig. 5C, D)[58]. Interestingly, Cox-2 overexpression also significantly suppressed T cell, NK cell, and Langerhans cell presence (Supplementary Fig. 5E), similar to the immunosuppressive functions of Cyclooxygenase activity demonstrated in some cancer tissues[59,60].

To define how McSC migration rate changes in *RP* mice, *RP* and control animals were provided doxycycline one day prior to UVB, followed by the irradiation protocol detailed above. Dorsal skin was collected, sectioned, and stained for Dopachrome Tautomerase (Dct), a label for McSCs and differentiated melanocytes. As expected, *RP* mice showed significantly higher epidermal melanocyte translocation after UVB irradiation, in both male and female groups (Fig. 5E). Additionally, McSC migration was not detected in non-UVB irradiated *RP* skin (Supplementary Fig. 5F). To visualize melanocytes in whole tissue, we crossed *RP* mice with *Tyr-CreER, lsl-tdTomato* (*TT*) animals. A whole mount view of *RP-TT* skin showed significantly higher epidermal melanocytes compared to wildtype *TT* control mice, consistent with the results from quantified sections (Fig. 5F).

To determine if increased McSC migration in *RP* mice was due to a higher McSC proliferation rate, we administered EdU to *RP-TT* mice four times per day from the second UVB until two days following the third UVB exposure, thereby labeling all proliferating cells within this time window (Fig. 5G). Click-iT™ staining showed higher EdU colocalization with tdTomato+ McSCs in *RP* skin compared to controls, indicating that increased epidermal melanocytes in *RP-TT* mice is due to increased proliferation from McSCs originally found in hair follicles (Fig. 5H). Moreover, the majority of epidermal melanocytes in both control and *RP-TT* mice were EdU+ (Ctrl: 75% and RP: 97%), indicating that proliferation is preferred prior to migration to the epidermis (Supplementary Fig. 5G). Given these data, we conclude that overexpression of Cox-2 heightens the inflammatory responses induced by UVB irradiation, which significantly promotes McSC proliferation and migration in both genders.

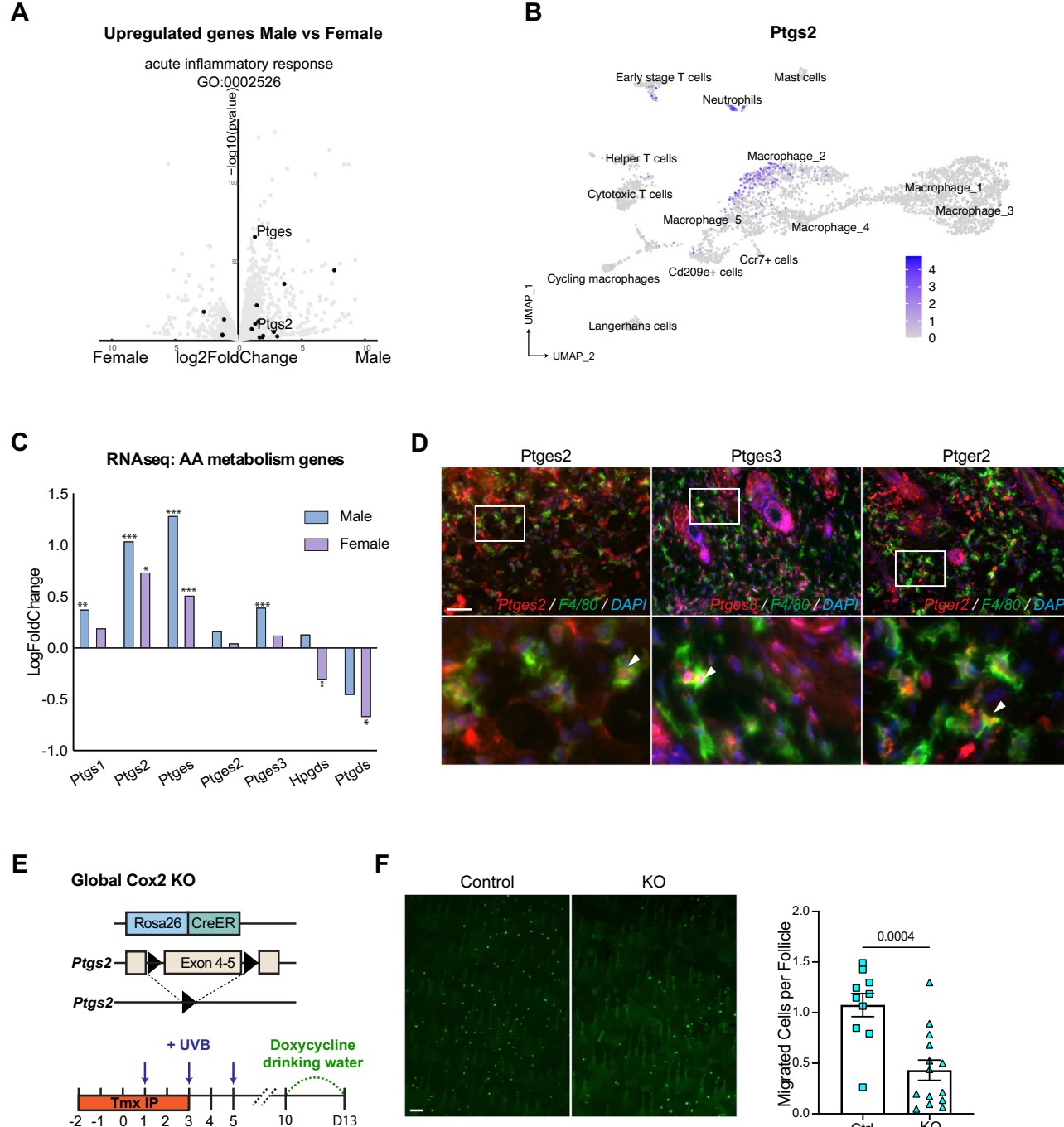

**Fig. 4 | Enhanced prostaglandin signaling in males regulates McSC migration.**
**A** Volcano plot of differentially upregulated genes in female (left) and male (right) mice. Genes that belong to Acute Inflammatory Response are highlighted.
**B** Expression level of *Ptgs2* in all immune cell subclusters described in Fig. 3B. **C** Log-fold gene expression changes in cyclooxygenases and select prostaglandin synthesis factors following UVB irradiation in males and females, from bulk mRNA sequencing data. *, **, *** indicates p.adjusts (apeglm shrinkage) are lower than 0.05, 0.01, 0.001, respectively. **D** Antibody staining for Ptges2, Ptges3, Ptger2, co-stained with F4/80 and DAPI on control skin collected at day 3. Co-localizations are indicated by circle. *n* = 3 males. **E** Schematic of *Rosa26-CreER; Ptgs2^{f/f}* mouse model, created with BioRender.com. **F** Representative images of McSC migration and quantification from Cox-2 global knockout mice compared to controls. *n* = 10 male mice for control (Ctrl), *n* = 14 male for knockout (KO). *Ptgs2^{f/+}* or *Ptgs2^{+/+}* were used as controls. Statistics: significance calculated by Welch's *t* test. Error bar: SEM. Scale bar: 100 μm.

In *RP* mice, Cox-2 overexpression was mainly found in epidermal keratinocytes and infiltrated immune cells. From scRNA-seq data, neutrophils and macrophages express relatively high *Ptgs2* within the immune cell population. Previous work has shown that UV is sufficient to upregulate *Ptgs2* expression in keratinocytes[53]. To determine if Cox-2 overexpression in specific cell types is sufficient to increase McSC migration, we utilized *Krt5-CreER; lsl-rtTA3; Tre-Ptgs2* (KLP) and *Lysm-Cre, lsl-rtTA3; Tre-Ptgs2* (MLP) mice to conditionally overexpress Cox-2 in keratinocytes and monocytes through doxycycline administration (Fig. 5I, K). Cox-2 overexpression in keratinocytes significantly increased McSC migration (Fig. 5J), while Cox-2 overexpression in Lysm+ monocytes and macrophages increased McSC migration, but with a lower fold change (Fig. 5L). However, Cox-2 IF staining on both genotypes showed a significantly different Cox-2 overexpression level

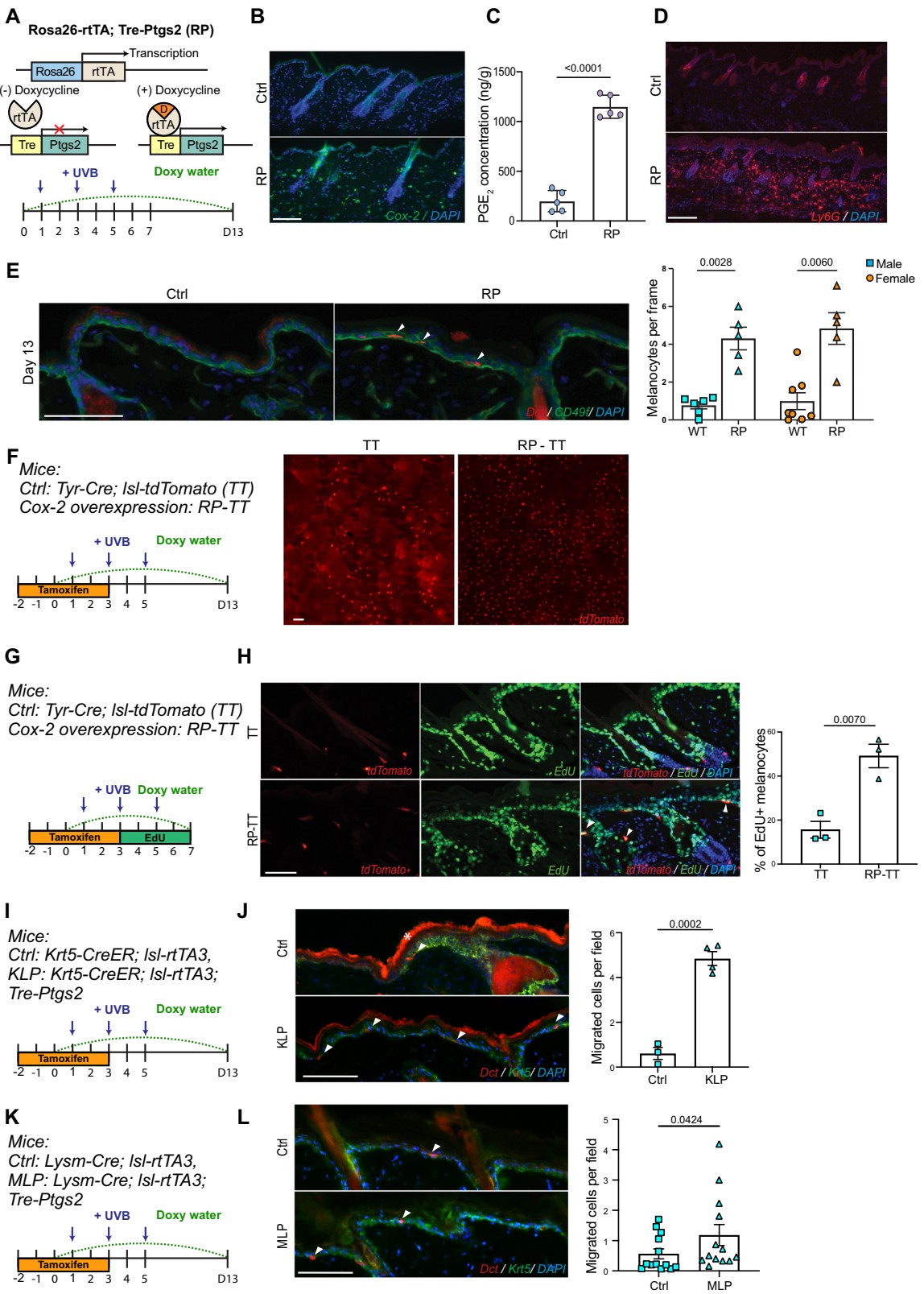

in the two mouse models (Supplementary Fig. 5H). Therefore, we cannot rule out the possibility that the different McSC migration efficiency is caused by efficiency variation in each Cre recombinase allele. Considering *Ptgs2* are expressed by almost all cell types and PGE$_2$ signals to downstream EP receptors through both autocrine and paracrine manners, it is possible that Cox-2 signaling is dosage dependent rather than cell type-specific.

## A distinct hybrid macrophage population is found in Cox-2 overexpression mice

To investigate how Cox-2 overexpression alters the inflammatory response to UVB irradiation, we profiled the transcriptomes of CD45+ immune cells from UVB irradiated *RP* skin collected at day 5, the same timepoint as the WT_UV single cell sample (Fig. 3A). After data integration, we found that there was increased neutrophil infiltration, and

**Fig. 5 | Increased cyclooxygenase expression escalates McSC proliferation and epidermal repopulation. A** Schematic of *Rosa26-rtTA; Tre-Ptgs2* (RP) mouse model (top), created with BioRender.com. Doxycycline administration and UVB irradiation timeline (bottom). **B** Representative Cox-2 staining on Ctrl and RP skin on day 2. $n = 3$ male Ctrl, 3 male RP, 3 female Ctrl, 3 female RP. **C** ELISA quantification of PGE$_2$ extracted from the UVB irradiated skin area, collected at day 5. PGE$_2$ level was normalized by the tissue weight. $n = 5$ mice for Ctrl and $n = 5$ mice for RP. **D** Representative Ly6G staining on Ctrl and RP skin collected on day 5. $n = 3$ mice for Ctrl and $n = 3$ mice for RP. **E** Representative images of Dct staining from Ctrl and RP skin collected on day 13 (left) and quantification (right). CD49f staining indicates the boundary of epidermis and dermis (left). Ctrl: 6m,8f; RP:5 m,5f. Migration rate was calculated by the number of epidermal melanocytes per frame. **F** Schematic of McSC labeling and doxycycline induction timeline in RP mouse model crossed with *Tyr-CreER, lsl-tdTomato* mice (TT, left). Representative images of Ctrl and RP whole-mount skin collected on day 13 (right). Each tdTomato-labeled red cell indicates one migrated melanocyte. $n = 4$ (2 TT, 2 RP-TT) (**G**) Timeline of melanocyte proliferation assay on TT and RP-TT mice. EdU was provided by I.P. injection every 6hs from day 3 until day 5. **H** Representative images of tdTomato melanocytes co-labeled with EdU by Click-iT reaction (left), and quantification (right). Arrows indicate the colocalized cells. Male: $n = 3$ mice for Ctrl, 3 mice for RP. **I, K** Timeline of Cox2 overexpression induction and melanocyte migration assay in *KLP* and *MLP* mice. **J, L** Representative images of migrated melanocytes and quantification. Arrowheads mark epidermal melanocytes. * marks autofluorescence from the stratum corneum. Male: $n = 3$ for Ctrl, $n = 4$ for *KLP*. Male: $n = 8$ pairs; Female: $n = 5$ pairs for *MLP*. *Rosa26-rtTA* allele-only or *Tre-Ptges2* allele-only were used as controls. Statistics: each individual dot in all panels indicates the averaged quantification from one mouse (**C, E, H, J**), with significance calculated by Welch's $t$ test. In (**L**), significance was calculated by paired $t$-test. Error bar: SEM. Scale bar: 100 $\mu$m.

fewer T cells and Langerhans cells in *RP* skin compared to the UVB-irradiated control sample (Fig. 6A, B), consistent with immuno-fluorescent staining results (Fig. 5D, Supplementary Fig. 5E). PGE$_2$ synthases, as well as PGE$_2$ receptors, are known to be expressed by macrophages and PGE$_2$ has been shown to modulate macrophage activates[51,61]. To better define the macrophage signatures in *RP* mice, we subclustered the macrophage populations from all three samples and labeled the clusters based on previously established marker genes[48]. *Retnla* and *Ear2* double positive clusters were defined as early-stage cells with high levels of the complement cascade, and alternative activation gene-expressed clusters were defined as the final stage of phagocytic macrophages (Supplementary Fig. 6A). Cell clusters in the middle were defined as Early, Intermediate, or Late-stage macrophages (Supplementary Fig. 6B). Very surprisingly, in addition to cycling macrophages and initial-stage macrophages being shared by all three samples, we found two unique macrophage subpopulations in the *RP* sample that did not fit into either the inflammatory path or phagocytic path (Fig. 6C).

To determine if *RP* macrophages contributed to heightened inflammation, we examined the expression of an inflammation gene set, including *Il1b, Tnf, Cxcl2, Ccl2, Jun, Junb, Fos*, and *Nfkbia*. As expected, macrophages showing the inflammatory path were identified in the control with UVB samples, and both the Intermediate_RP and Final_RP populations expressed these inflammatory genes as well. Consistent with this, the intermediate stage- and final stage macrophages in the control without UVB sample showed low expression of the genes in this set (Fig. 6D, E). Moreover, the pha-gocytotic gene set[48] was plotted and surprisingly, we found that the Intermediate_RP and Final_RP macrophage subpopulations exhibited both inflammatory and phagocytotic signatures (Fig. 6F), challenging the concept of four diverse activation paths[48]. From these data, we conclude that in an environment of Cox-2 overexpression, distinct from the two separate activation paths found in control samples, induction of both inflammation (inflammatory path) and resolving (phagocytotic path) signatures are present in the same group of macrophages.

To determine the necessity of macrophages with increased Cox-2 in McSC migration, *RP* mice were treated with clodronate liposomes (CL) to deplete circulating monocytes. Flow cytometry and immuno-fluorescent staining showed CL-treated mice had significantly decreased monocytes in blood and in irradiated skin tissue compared with *RP* mice treated with PBS-containing liposomes (PL). To determine the effect of macrophage depletion on McSC migration, we collected CL and PL-treated *RP* mice and stained epidermal melanocytes with an anti-Dct antibody (Fig. 6G). Immunofluorescence staining showed a significant decrease in McSC migration in the CL depletion group compared to the PL-treated cohort (Fig. 6H). To understand if the decrease in McSC migration is due to impaired McSC proliferation or migration, we labeled all proliferating cells with EdU in *RP-TT* mice while treating with CL. CL treatment resulted in a significantly decreased McSC proliferation rate in *RP* mice (Fig. 6I, J). In conclusion, macrophage depletion in *RP* mice results in decreased McSC migration, consistent with our data showing that DT-mediated macrophage depletion in wild-type mice exhibits a similar migration reduction.

## Both dmPGE$_2$ supplementation and dmPGE$_2$ with Jak signaling inhibition promote UVB-induced McSC epidermal repopulation

Increased melanocyte repopulation is needed to improve the scale of repigmentation in patients with depigmentation conditions. To determine whether cyclooxygenase products could hold therapeutic potential in promoting hair follicle McSC colonization of the epidermis, we administrated 16,16-dimethyl Prostaglandin E$_2$ (dmPGE$_2$, a stabilized, synthetic form of PGE$_2$) to murine dorsal skin in areas of UVB irradiation[62–64]. dmPGE$_2$ treated mice showed significantly increased McSC migration in both males and females after UVB irradiation (Fig. 7A, B), while dmPGE$_2$ without UVB was not sufficient to induce McSC migration (Supplementary Fig. 7A). This suggests that PGE$_2$ application during phototherapy in human patients may result in improved outcomes. This result also suggests that cell-specific Cox-2 and PGE$_2$ are not necessary to induce McSC migration. Interestingly, dmPGE$_2$ application significantly promoted neutrophil infiltration while reducing T cell recruitment (Fig. 7C, D), similar to the pro-inflammation phenotype found in *RP* mice (Fig. 5D, Supplementary Fig. 5E).

Recently, phase II and III clinical trials with Jak inhibition have shown positive results for skin repigmentation in vitiligo patients[8,9], resulting in FDA approval of ruxolitinib[65]. Jak inhibitors such as rux-olitinib and tofacitinib prevent the recruitment of CD8$^+$ T cells to lesional skin and limit cytotoxic T cell-mediated destruction of epidermal melanocytes[6]. However, the effectiveness of ruxolitinib is still limited, with only around 20% of patients achieving 50% repigmentation of the total lesion area (T-VASI50) after 52 weeks of treatment. Interestingly, ruxolitinib treatment achieved higher facial re-pigmentation (50% of T-VASI50) than total body re-pigmentation (20% of T-VASI50), suggesting exposure to environmental UVB in this body region has a synergistic or additive effect for repigmentation[9]. Using our mouse model, the JAK inhibitor ruxolitinib, without UVB exposure, did not induce obvious epidermal migration under short-term treatment (Supplementary Fig. 7B). This data indicates that the combination of a JAK inhibitor with UVB irradiation may be a more effective repigmentation method, consistent with a recent small sample study which reported that UVB irradiation improved the repigmentation effect of ruxolitinib in patients[66].

Topical PGE$_2$ gel is approved for cervical ripening in human patients[67], and small clinical reports have shown the safety and potential positive effect of PGE$_2$ on skin re-pigmentation[68,69]. To test a therapy that utilizes inflammatory pathways to promote re-pigmentation while not interfering with the immunosuppressant activity of JAK inhibitors, we systemically treated with ruxolitinib while providing dmPGE$_2$ intradermally to UVB-irradiated *DG* mice,

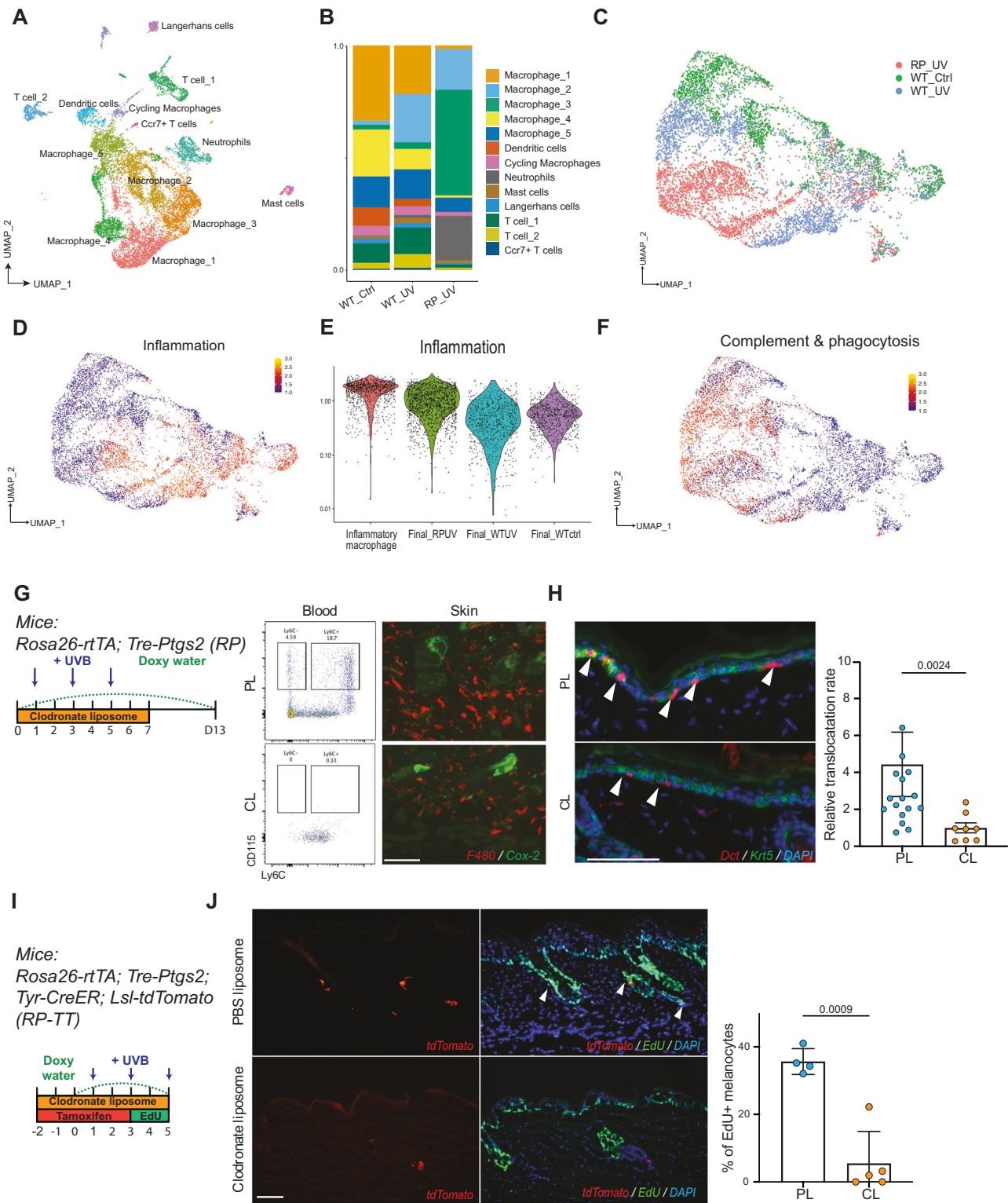

with ruxolitinib-only treated *DG* mice used as controls (Fig. 7E). Ruxolitinib did not suppress the migration effect of dmPGE$_2$, but instead further promoted McSC translocation to the IFE (Fig. 7F). Further investigation is needed to define the mechanism through which JAK inhibition cooperates with UVB exposure supplemented with dmPGE2. Together, our data suggests that combined dmPGE$_2$ and ruxolitinib promote UVB-induced McSC migration, which could be translationally beneficial for improving and maintaining repigmentation in vitiligo patches.

## Discussion

Extensive studies on vitiligo have focused on T cell suppression, and Jak inhibitors to suppress T cell recruitment have been approved for vitiligo therapy[6,7]. Even though significant amounts of repigmentation is often achieved from this treatment, efficacy is still limited and varies among patients[8,9]. Previous studies have suggested that achieving durable repigmentation will likely require combinatorial approaches that address both vitiligo pathogenesis and epidermal melanocyte repopulation[5]. In this study, we aimed to increase the level of

**Fig. 6 | A distinct hybrid macrophage population is found in Cox-2 overexpression mice. A** UMAP of immune cell clusters from WT_Ctrl, WT_UV, and RP_UV. WT_Ctrl and WT_UV were the same dataset from Fig. 3. RP_UV sample was from an RP male mouse collected on day 5, the same timepoint as WT_UV. **B** Barplot showing the composition of each cell cluster within the three samples. **C** UMAP of macrophage subclusters isolated from all three samples. Macrophages were sub-clustered from cluster Macrophage_1–4 and Proliferative macrophages. **D** Visualization of the Inflammation gene set expression in macrophage subclusters. **E** Violin plot showing expression of the Inflammation gene set in Inflammatory macrophage, Final_RPUV, Final_WTUV, Final_WTCtrl subclusters. **F** Visualization of Complement & Phagocytosis gene set expression in macrophage subclusters. **G** Schematic timeline for macrophage depletion using clodronate liposomes (CL), and control PBS liposomes (PL). Monocyte depletion in the

circulation was validated by flow cytometry. Blood monocytes were identified as CD45 + CD115+. Ly6C was used to classify infiltrating monocytes (Ly6C$^{hi}$). Reduced skin macrophage infiltration by CL depletion was validated by F4/80 staining on sectioned skin collected on day 5. **H** Final melanocyte migrations were quantified by Dct staining on sectioned skin collected on day 13. Migrations were normalized based on individual level of Cox-2 expression in each mouse. $n = 8$ mice for CL (4 male, 4 female), $n = 17$ for PL (9 male, 8 female). Uneven number due to the poor animal survival under CL treatment. **I** Schematic of melanocyte proliferation assay on RP-TT mice under macrophage depletion. Doxycycline, tamoxifen, and PL/CL treatments follow the timeline. **J** Representative images of tdTomato co-labeling with EdU staining (left), and quantification (right). $n = 5$ mice for CL, $n = 4$ mice for PL. Scale bar: 100 μm. Statistics: Mann-Whitney $t$ test (**H**), Welch's $t$ test (**J**). Error bar: SEM. Each individual dot indicates the averaged quantification from one mouse.

epidermal melanocyte translocation that normally occurs by phototherapy-induced migration of hair follicle McSCs. We found that the UVB-induced inflammatory response underlies McSC migration to the epidermis and that the gender-specific differences in inflammation levels translate to differences in McSC translocation rates. We also demonstrated that applying supplemental dmPGE2 promotes UVB-induced inflammation and further addition of JAK inhibitor treatment results in a significantly higher number of epidermal melanocytes than any single therapy alone.

Many studies have demonstrated sex differences in gene regulation, disease incidence, and therapeutic efficacy[32,70]. Recently, increased reporting of sex and gender in research has provided a better understanding of experimental results and has aided in reproducibility[71]. Therefore, to define any dynamic differences in McSC response to UVB between sex, we treated each gender as a separate cohorts. Surprisingly, we found that male mice show a significantly higher McSC migration rate than females (Fig. 2A). While there is currently a lack of reports utilizing well-controlled, sex-specific cohorts to assess repigmentation efficiency via phototherapy, our data implies that male vitiligo patients may experience better repigmentation outcomes than female vitiligo patients. It is crucial to note that various factors, including melanin production, melanin transportation and epidermal melanocyte maintenance, affect the repigmentation process. Nevertheless, it would be worthwhile to explore this hypothesis in further clinical investigations.

Both RNAseq data and flow cytometry results indicated that male mice experience a significantly higher inflammatory response than females under the same UVB irradiation dosage (Fig. 2E, G). Inflammatory response differences between genders have been reported in both humans and rodents under different pathogenic or tissue damage conditions[72,73]. With respect to UVB-induced damage, we found that cyclooxygenases and downstream prostaglandin E2 synthases were highly upregulated in males (Fig. 4C). To determine if prostaglandin signaling is responsible for the increased inflammatory response and McSC migration in males, we manipulated the Cox-2 coding gene *Ptgs2* by global knockout and overexpression (Figs. 4E, 5A). As predicted, knocking out Cox-2 reduced McSC migration, while overexpressing Cox-2 dramatically increased migration. Furthermore, cell type-specific Cox-2 overexpression and extrinsic dmPGE2 administration increased migration as well (Figs. 5J, L, 7B). Even though male mice expressed higher *Ptgs2* and induce an increased inflammatory response, the upstream signals that drive higher expression of *Ptgs2* in males are still unclear. Transcription factor AP-1 has been shown to regulate *Ptgs2* mRNA expression[74], and AP-1 activities can be regulated by sex hormones through estrogen receptor α[75]. Sex hormones have been implicated in gene expression regulation and can cause different phenotypic outputs in human diseases[70]. It will be tantalizing to investigate the relationships between sex hormones and *Ptgs2* expression under UVB irradiation in future studies.

In our non-autoimmune mouse models, Cox-2 overexpression (Supplementary Fig. 5E) and dmPGE2 application (Fig. 7D) limited

T-cell infiltration in UVB-irradiated skin tissue. Since we did not induce CD8 + T cell cytotoxicity to melanocytes in our models, it is very unlikely that the increased epidermal melanocyte repopulation under these treatments was due to T cell suppression. Instead, studies on the tumor microenvironment show that high Cox-2 and prostaglandin E2 levels in tumors suppress T cell infiltration and lead to worse prognosis[59,60,76], which is consistent with the limited T cell infiltration in Cox-2-overexpressed and dmPGE$_2$-treated skin tissue shown in this study. Here, we demonstrated the pro-inflammatory role of macrophages and its impact on McSC epidermal migration following UVB irradiation. Inflammation is a well-studied biological process that involves a broad range of cells and molecules, including but not limited to macrophages, neutrophils, dendritic cells[77], T cells[78], keratinocytes[79], and fibroblasts[80]. Future investigations may shed light on the roles of other cell types and pathways in regulating the inflammatory response in the skin after UVB irradiation. It should be noted that McSC migration was not induced independently by imiquimod, dmPGE$_2$ or Cox-2 overexpression in the absence of UVB irradiation, suggesting that other cell types and signaling pathways are involved in McSC migration, in addition to inflammation, such as Kitl expression in keratinocytes under UVB expression[15].

In summary, we discovered that male mice show a higher inflammatory response following UVB irradiation compared to female mice, which causes enhanced McSC migration. By over-expressing the pro-inflammatory factor Cox-2, applying the pro-inflammation agent imiquimod, or treating directly with dmPGE$_2$, we successfully increased UVB-induced McSC migration. Two requirements for successful re-pigmentation include immunosuppression and epidermal melanocyte repopulation[5,81]. Thus, we tested the immunosuppressant drug ruxolitinib in combination with dmPGE2, which showed an increase in efficacy (Fig. 7E, F). Taken together, this work demonstrates the translational feasibility of combining pro-inflammation modalities (to promote enhanced melanocyte recruitment) with anti-T cell agents to generate a more favorable outcome compared to phototherapy and immunosuppressant-only treatments used currently.

## Methods

### Mice

*Rosa26-CreER* (Jax, stock #008463), *Ptgs2$^{flox/flox}$* (Jax, stock #030785), *Rosa26-rtTA* (Jax, stock #029627), *Tre-Ptgs2* (Jax, stock #033884), *Lgr6-CreER*[30] (Jax, stock #016934), *Lysm-Cre*[82] (Jax, stock #004781), *Rosa26-lsl-rtTA3*[83] (Jax, stock #029617), *Rosa26-lsl-iDtr*[84] (Jax, stock #007900) were purchased from The Jackson Laboratory. NSG mice were obtained from Dr. Rishi Puri at Cornell University. *Dct-rtTA; TRE-H2B-Gfp* on the C57Bl/6N background was provided by Dr. Glenn Merlino at the National Cancer Institute. All mouse lines were crossed and maintained on C57Bl/6N or C57Bl/6J backgrounds. Unless otherwise noted, McSC migration assessments using the *Dct-rtTA* system were performed on a C57Bl/6N background. All mouse treatments were in

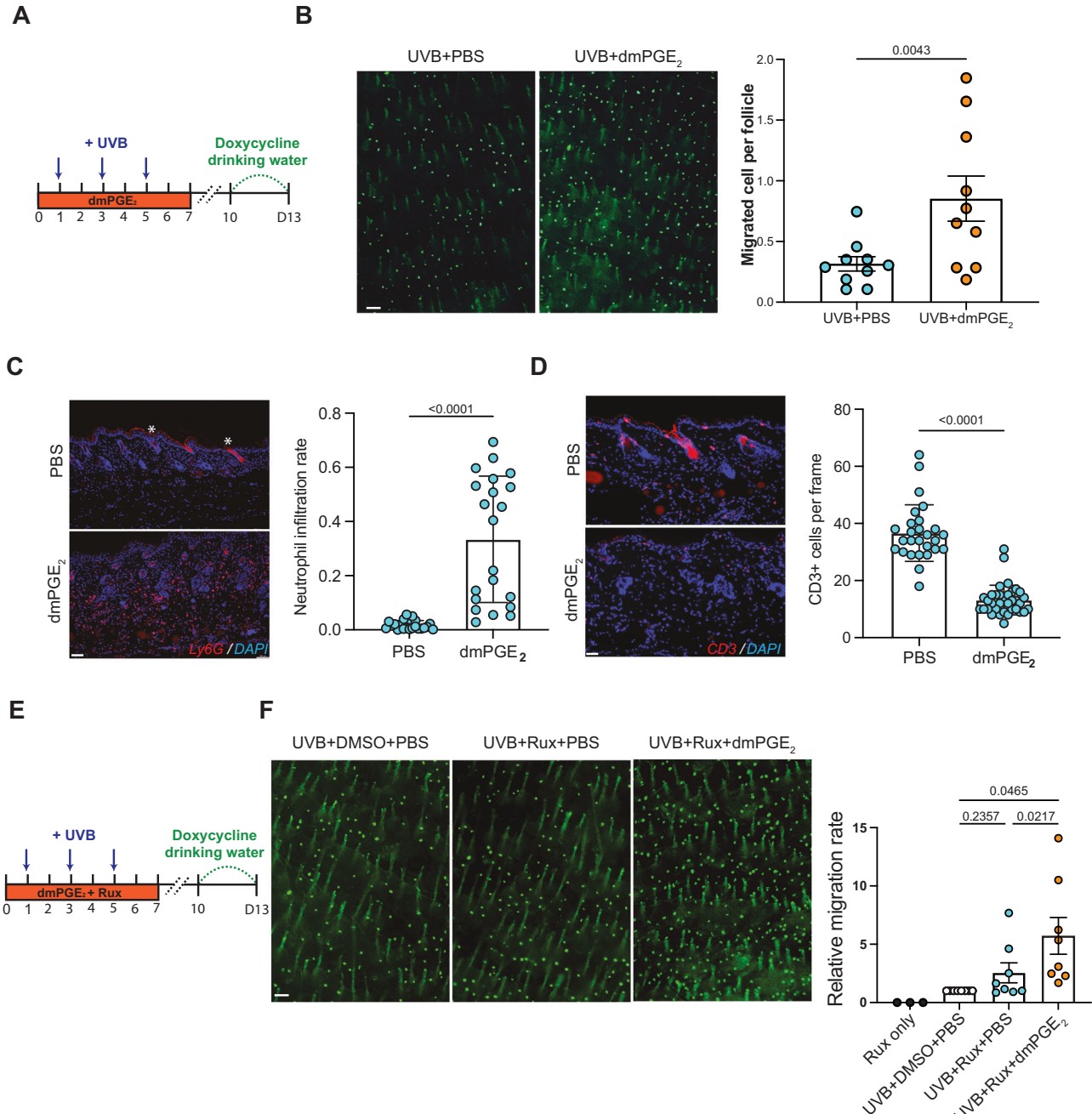

**Fig. 7 | Both dmPGE$_2$ supplementation and dmPGE2 with Jak signaling inhibition promote UVB-induced McSC epidermal repopulation. A** Schematic timeline of dmPGE$_2$ treatment by intradermal injection and UVB irradiation.
**B** Representative images of whole mount DG skin collected at day 13 (left), and migration quantification (right). Migrated melanocytes from at least 6000 follicles were quantified in each mouse. $n = 8$ pairs (4 male pairs, 4 female pairs). Paired comparisons were performed by using littermates to minimize the effect of genetic variance and to minimize any UVB light variance. **C, D** Representative images of neutrophil infiltration (Ly6G+) and T cell infiltration (CD3+) in UVB irradiated PBS-treated skin collected on day 4. Red staining in the PBS panel is background in the stratum corneum and hair shaft (labeled by *). Neutrophil infiltration rate was quantified by the Ly6G+ staining area over DAPI staining area, and CD3 + T cell infiltration was quantified by the number of T cells per image. Each individual dot indicates one image. Ly6G: $n = 22$ images for PBS, $n = 21$ images for dmPGE$_2$; CD3: $n = 27$ images for PBS, $n = 35$ images for dmPGE$_2$. Each group contains $n = 3$ male mice. **E** Schematic of dmPGE$_2$ and ruxolitinib administration with UVB irradiation. **F** Representative whole mount images of DG dorsal skin collected on day 13 (left), and quantification. For each group of experiments, three littermates of the same gender were used for UVB-only; UVB+ruxolitinib; and UVB+ruxolitinib+dmPGE2 treatments. To normalize the variance of migration rate due to different experiment date and gender, the migration rate of UVB+ruxolitinib+PBS and UVB+rux-olitinib+dmPGE2 animals were normalized by the migration rate in UVB + DMSO + PBS control individuals within each group. Three ruxolitinib-treated non-UVB irradiated males were used to show no migration without UVB. Scale bar: 100 μm. Statistics: Paired-$t$-test (**B**, **F**), Welch's $t$-test (**C**, **D**). Error bar: SEM.

line with the protocol "2014-0096" approved by the Institutional Animal Care and Use Committee (IACUC) at Cornell University. Mice were maintained under pathogen-free conditions at Cornell University College of Veterinary Medicine by the Center of Animal Resources and Education (CARE). All genotyping was performed as previously published or as described by the Jackson Labs[20]. All melanocyte migration experiments were done on telogen dorsal skin beginning at 7–9 weeks of age in both genders.

## Melanocyte labeling system

For *Dct-rtTA; Tre-H2B-Gfp* mice (NCI, stock #01XT4), 200 µg/ml doxycycline (Alfa Aesar, Cat#J60579) in drinking water was provided 3 days before collection to induce H2B-GFP labeling. For *Tyr-CreER; LSL-tdTomato* animals (Jax, Stock# 012328, Stock #007908), tamoxifen (Cayman, Cat#13258) was dissolved in Corn Oil with 10% ethanol at 20 µg/ml, and 200 µl was injected three days prior to the first UVB irradiation for five consecutive days to label with tdTomato.

## Skin UVB irradiation and acquisition

The UVB irradiation procedure was performed as previously published[85]. Briefly, before irradiation, the UV lamp (Analytikjena, Part# 95-0042-03) was warmed up to reach a consistent intensity, as monitored by a UV Radiometer (Analytikjena, Part# 97-0015-02, Part# 97-0016-04). Shaved mice were mounted in the UV lamp apparatus, covered by a 302/26 nm single-band bandpass filter (AVR Optics, Part#FF01-302/26) and irradiated for 40 s under anesthesia. Each UVB irradiation timeline is specified in the corresponding Figure and Figure legend. Seven days following the third UVB irradiation, mice were euthanized, and the irradiated skin was shaved using shave gel (Gillette) and razor (Up&Up). A 1.2 cm × 1.0 cm skin region in the center of the irradiated area was trimmed and mounted on a slide for whole-mount imaging. Images were taken using a Leica DM7200 fluorescence imaging platform with LAS X software version 3.7.5. In general, 20–25 images were taken for each mouse, and around 320 hair follicles were in each image. Thus, for each mouse, migrated melanocytes from 6400 to 8000 hair follicles were quantified and normalized.

## Whole-mount skin imaging and McSC migration quantification

Images were taken in both green and red channels. Autofluorescence from hair follicles in the red channel was used to quantify the number of hair follicles in the field. Red channel autofluorescence was used to balance the same autofluorescence in green channels and highlight H2B-GFP labeled McSCs in the epidermis. The number of H2B-GFP points in each field was quantified as the total number of migrated McSCs. McSC migration rate was calculated by the number of migrated McSCs per the number of hair follicles. *Dct-rtTA; Tre-H2B-Gfp* mice on a C57Bl/6J background showed fewer migrated McSCs than mice on a C57Bl/6N background. All image quantifications were performed by a program created in MATLAB and corrected by eye. Code can be found on figshare.

$$\text{Migration rate} = \frac{\text{number of McSCs}}{\text{number of hair follicles}}$$

## Two-photon live imaging

Mice were anesthetized with 4% isoflurane in oxygen, maintained at 1.8–2%, placed on a feedback-controlled heating pad to maintain body temperature at 37 °C (Harvard Apparatus) and mounted on a custom-designed stereotaxic instrument for back-imaging. Before imaging, mice received either glycopyrrolate (0.05 mg/100 g of mouse body-weight, Hikma Pharmaceuticals) or atropine sulfate (0.005 mg/100 g of mouse bodyweight, Med-Pharmex Inc.) subcutaneously to keep the airways clear of fluid build-up. Eyes were covered with veterinary eye ointment to prevent drying. The mice were hydrated with subcutaneous injections of 5% glucose in saline. The dorsal skin was shaved as mentioned previously. To help relocate the same location, the region of interest (ROI) was labeled by tattoo ink. Tattoo ink (Ketchum, Cat#329AA) was injected into the shaved mouse skin (on the edge of irradiated region to minimize wounding effects) with a total amount of 400 nl at a depth of 50 um, at a speed of 50 nl/s with a pressurized injector (Drummond, Nanoject III) controlled with micromanipulators (Luigs and Neumann, SM-7; Scientifica PatchStar). Imaging was performed on a custom-made laser scanning two photon microscope run by SCANIMAGE. A femtosecond Yb:fiber (Satsuma,

Amplitude Systèmes) at 1030 nm wavelength was used for this experiment. A 4×, 0.28 Numerical Aperture (NA) Objective (Olympus) was used for locating the ROI marked by tattoo while a 20×, 1.0 NA water immersion Objective (Zeiss) was then used for laser excitation, and collection of the fluorescence signal, which was epi-detected. Fluorescence was collected in four detection channels of the microscope formed by three long-pass dichroics: FF520-Di02, LM01-466, FF593-Di03. In addition, the channels had the following bandpass filters respectively, FF01-439/154, FF01-520/60, FF01-550/88, and FF01-615/45. The stack consisted of slices taken at 1 µm intervals from the dermal surface to a depth of 150–180 µm. After acquiring the images, the 20× objective was switched to 4× and the stereotaxic station was moved to locate the tattoo spot. Then, the relative coordinates between the tattoo spot and the image acquisition spot were recorded. The mouse was then taken from the microscope, allowed to recover from anesthesia, and was imaged 24 h later repeating the above procedure. To re-visit the same follicles, the 4× objective was first used to relocate the region labeled with tattoo ink and then switched to the 20× objective. Next, the station was moved based on the previously recorded coordinates, and the hair follicle pattern was visually confirmed. The imaging analysis was performed with ImageJ (NIH) and Imaris 9.0 (Oxford Instruments) software.

## Cox-2 knockout system

For *Rosa26-CreER; Ptgs2^{ff}* mice, 200 µl of tamoxifen (Cayman, Cat#13258) at 20 µg/ml was injected 3 days before the first UVB exposure, and for 5 consecutive days following, to induce sufficient knockout.

## EdU cell proliferation assay

5-ethynyl-2′-deoxyuridine (Cayman, Cat#20518) was dissolved in PBS with 10% ethanol at a concentration of 3.33 mg/ml. 100 µl of this EdU solution was intraperitoneally injected four times per day. After mice were collected, skin was embedded in OCT (Fisher, Cat#23730571) blocks and sectioned at 8 µm thickness. Stainings were performed following the protocol in the Click-iT EdU Cell Proliferation Kit manual (ThermoFisher, Cat#C10640, Cat#C10637).

## Immunofluorescent staining and imaging

Dorsal skin samples were embedded in OCT and sectioned at 8 µm thickness. Sections were fixed in formalin for 10 min, followed by two $H_2O$ washes (10 min each). Sections were blocked with blocking buffer (10% donkey serum in 1xPBST) for 1 h. Primary antibodies were diluted in blocking buffer and applied to the tissue and incubated overnight at 4 °C. Secondary antibodies were diluted in 1xPBST for 1 h at room temperature following three 1xPBST washes (5 min). Sections were washed in 1xPBST three times (5 min) and mounted with Fluoroshield with DAPI (Abcam, ab104139). Images were taken using the Leica DM7200 fluorescence imaging platform with LAS X version 3.7.5. Antibodies used: F4/80 1:600 (Biolegend, Cat#123101); Ly6G 1:600 (Biolegend, Cat#127601); CD3 1:800 (Biolegend, Cat#100201); Cox-2 1:600 (Cayman, Cat#160106); Ptges2 1:200 (Abclonal, Cat#A7137); Ptges3 1:200 (Abclonal, Cat#A5194); Ptger2 1:500 (Abcam, Cat#ab167171); Cd49f 1:100 (BD Biosciences, Cat#555734); Dct 1:600 (Abcam, Cat#ab221144), together with suitable Alex Fluor secondary antibodies (Abcam, Cat#ab150072, ab150149, ab150152, Fisher, Cat#A21207)

## Bulk RNA sequencing

Whole UVB irradiated skin samples were collected and homogenized under liquid nitrogen. Whole RNA was isolated from tissue powder using TRIzol™ (Life Technologies, Cat#10296028). 500 ng of total RNA was used as the library prep input. Poly(A) mRNA purification Module (NEB, Cat#E7490S) and the NEBNext Ultra II directional RNA library Prep kit (NEB, Cat#E7785S) for Illumina were used for index

library prep following NEB #E7760S/L, #E7765S/L version 4.0_4/21. The NextSeq 2 K P2 100 bp kit platform was used for sequencing. Sequencing reads were aligned to the mouse genome using STAR. Differential expression analysis was performed using DESeq2. Statistical significance was given to genes using adjusted *P* value of 0.1 according to the Benjamini-Hochberg adjustment, and shrinked log fold change larger than 1 according to "apeglm" shrinkage method. Gene Ontology analysis was performed using "clusterprofiler" package.

## Single-cell RNA sequencing

Single-cell gene expression profiles were generated using the 10x Genomics Chromium (3′RNA-seq) platform. Briefly, skin tissues were digested using 20 mg/mL Collagenase (Worthington, Cat#LS004196) dissolved in DMEM/F12 (Corning, VWR# 45000-350) media. Digested tissues were passed through 100 and 70 μm nylon cell strainers (Corning, VWR# 10054-458, 89508-344). Single-cell suspensions were washed by MACS buffer, incubated with CD45+ magnetic microbeads (Miltenyi Biotec, Cat# 130-052-301), and passed through MS Columns (Miltenyi Biotec, 130-042-201) following the protocol provided in the kit. MACS buffer consists of 1xPBS with 2 mM EDTA (VWR, Cat#97062-836) and 0.5% BSA (VWR, Cat#EM2930). Single-cell suspensions were kept in PBS with 0.04% BSA for sequencing. Sequencing results were processed by Cell Ranger. QC was analyzed by Seurat with percent.mt<10; nFeature_RNA > 200, nFeature_RNA < 4000, (WT_UV nFeatures_RNA < 6000 due to deeper illumina sequencing). Data integration was accomplished by Seurat4 SCTintegration. Pseudotime analysis was performed by Monocle 3. CellChat signaling network analysis was performed by CellChat 1.5.0. Code can be found in figshare.

## ELISA

PGE₂ and PGD₂ ELISA assays were performed using Cayman Prostaglandin E2 Express ELISA kit (Cayman, Cat#500141-96,4000020-25) and Cayman Prostaglandin D2 Express ELISA kit (Cayman, Cat#512041-96). Whole skin was used to perform prostaglandin extraction following the guidance of the protocol included in the kit. Prostaglandins were normalized by the weight of the tissue.

## Flow cytometry

Whole blood was collected via cardiac puncture from euthanized animals. Red blood cells were lysed using ACK lysis buffer (Fisher, Cat#A1049201), and then the remaining cells were stained with fluorochrome-conjugated antibodies. Skin tissues were digested as described in the protocol above. Single-cell suspensions were stained with fluorochrome-conjugated antibodies. Data were analyzed by FlowJo v10.8 (DB Life Sciences). The following antibodies and dyes were used: CD45 (1:200, clone 30-F11; BD Biosciences Cat# 564279), CD11b (1:200, clone M1/70; Thermo Fisher Scientific Cat# 45-0112-80), CD115 (1:200, clone AFS98; Thermo Fisher Scientific Cat# 12-1152-81), Ly6C (1:200, clone HK1.4; Thermo Fisher Scientific Cat# 47-5932-80), Ly6G (1:200, clone 1A8, BD Biosciences Cat# 565369), F4/80 (1:200, clone BM8, Thermo Fisher Scientific Cat# 17-4801-80), Live/Dead Aqua or Violet fixable stains (1:800, Life: L34964, NC0180395).

## Clodronate liposome treatment

Mice were given doxycycline in water and the first dose of clodronate liposomes (CL) or PBS liposomes (PL) (Liposoma, Cat#CP-005-005; Fisher, Cat# NC1438582) on the day prior to the first UVB irradiation. Male and female mice were given 250 μl and 200 μl CL/PL, respectively, due to weight differences. From one day after the first UVB, mice were given CL/PL daily until the collection day or 2 days following the third UVB. Mice for McSC translocation analysis were collected seven days following the third UVB irradiation.

## Imiquimod treatment

5% IMQ cream (Perrigo) or control Vanicream (Pharmaceutical Specialties Inc.) were applied topically on the UVB irradiated area with ~1 mg/cm² following the first and second UVB irradiation, as previously described[41].

## Prostaglandin treatment

10 μg of dmPGE₂ (Cayman Cat#14750) was dissolved in 50 μl PBS with 10% EtOH, and intradermally injected daily under UVB-irradiated skin.

## Diphtheria toxin treatment

Diphtheria toxin (Sigma, Cat#D0564) was dissolved in ddH₂O at a stock concentration of 600 ng/μl and stored at −80 °C. Before usage, stock solution was diluted with sterile 0.9% NaCl saline to a final concentration of 2.52 ng/μl. 200 μl, which was administered via I.P. injection.

## Ruxolitinib treatment

Ruxolitinib (MedChem Express, cat#HY-50856) was dissolved in DMSO for a stock concentration of 60 mg/ml. Prior to injection, 10 μl of this stock solution was dissolved in 200 μl saline. 10 μl DMSO dissolved in 200 μl saline was used in controls. 200 μl of the solution was delivered daily through oral gavage followed by dmPGE₂ by intradermal injection.

## Immunofluorescence staining quantification

Skin section images were taken on fluorescent and DAPI channels. Both fluorescent and DAPI staining were quantified by a program created in MATLAB and relative staining levels were calculated by the area of positive staining divided by the area of DAPI staining. Code can be found on figshare.

## Statistics

Except for bulk and single-cell RNA sequencing data, all statistical analyses were performed using Prism (GraphPad). All data are presented as mean ± SEM with two-sided comparision. Statistical details (including testing method, *N*, *p* values) are present in the corresponding Figure and Figure legend.

## Study approval

All animal procedures were approved by the Institutional Animal Care and Use Committee at Cornell University.

## Reporting summary

Further information on research design is available in the Nature Portfolio Reporting Summary linked to this article.

## Data availability

The raw and processed bulk and scRNA sequencing data generated in this study have been deposited in the GEO database under accession code GSE247532 and GSE247694 respectively. Source data are provided with this paper.

## Code availability

Code for image analysis, bulk mRNA sequencing, and scRNA seq are available in figshare "UVB-induced melanocyte stem cell activation".

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

## Acknowledgements

We thank Dr. Glenn Merlino for providing *Dct-rtta; TRE-H2B-GFP* mice backcrossed to C57Bl/6N. We thank Dr. John Harris and Laura Lajoie for their insights on vitiligo models. We thank Dr. Prashiela Manga, Dr. Paul Soloway, and Dr. Anushka Dongre for insights on the manuscript. We thank William Zhuang for genotyping; the Cornell Center for Animal Resources and Education (CARE) for maintaining mouse colonies; the Cornell Biotechnology Resource Center (BRC) Genomics Facility (RRID: SCR_021727), Bioinformatics Facility (RRID:SCR_021757), and Flow Cytometry Facility (RRID:SCR_021740) for processing our single cell and RNA samples and data, with special thanks to P. Schweitzer for providing technical support. This study was funded by the National Institutes of Health grant NIH 5R01AR075755, a Cornell Stem Cell Program seed grant, and a Cornell Center for Vertebrate Genomics scholar fellowship.

## Author contributions

Conceptualization: L.A., A.C.W. Methodology: L.A., D.K., M.A.M., C.Y.E., N.N. Investigation: L.A., D.K., L.D., N.N., A.C.W. Data Analysis: L.A., D.K. Funding acquisition: A.C.W. Project administration: A.C.W. Supervision: N.N., A.C.W. Writing—original draft: L.A., A.C.W. Writing—review & editing: L.A., D.K., and A.C.W.

## Competing interests

The corresponding author declares the following competing interests: Patent applicant: Cornell University. Name of the inventor: Andrew White. Application number: 63/496,967. Status of application: Pending. Specific aspect of manuscript covered in patent application: Figure 7, use of pge2 analog with uvb and uvb+ruxolitinib. The remaining authors declare no competing interests.
