## [Peer Review File · Nature Communications]

Sexual dimorphism in melanocyte stem cell behavior reveals combinational therapeutic strategies for cutaneous repigmentationREVIEWER COMMENTS

Reviewer #1 (Remarks to the Author):

The authors discovered that the migration rate of McSCs differs between male and female mice after UV treatment. Cox-2 and PGE2 were found to promote McSC proliferation and epidermal migration. The authors also identified a distinct subtype of macrophages that is involved in promoting MC migration. They proposed a working model that different inflammatory responses may explain the differences between males and females. The phenomenon is interesting, but the overall data quality needed to support their conclusions is not sufficient.

1. The author's results demonstrate increased MC migration and enhanced inflammatory response in male mice compared to female mice after UV radiation. Application of IMQ to both male and female mice resulted in increased MC migration. Therefore, the authors concluded there is a correlation between enhanced immune response and melanocyte migration. But UV treatment and IMQ treatment are two completely different things, using IMQ data to support immune cell difference is behind UV response difference between male and female is not logically coherent.

To investigate whether immune cells are involved in UV induced MC migration difference between male and female mice the authors could try using immune-deficient mice, to see whether it affects melanocyte migration.

2. The author's used a global Cre to knock out or overexpress *Ptgs2*, this cannot be used to draw conclusion of skin specific mechanisms. It is possible that the changes in melanocyte migration are caused by cells other than those residing in the skin. To support their conclusion, the authors need to use a skin cell-specific Cre mouse model or conduct skin graft experiments, for example the skin of mice with *Ptgs2* knockout/overexpression can be transplanted onto wild-type mice, and ask whether the phenotype maintains.

3. The author identified an inflammatory macrophage population in WT male mice after UV and in Cox-2 overexpression mice. However, in the case of Cox-2 overexpression mice, the author used clodronate liposomes to deplete all circulating macrophages without targeting this specific population of inflammatory macrophages, but the conclusions are drawn on the population of inflammatory macrophages. This conclusion need to be modified to reflect the experimental data. Without specific ablation experiment, it is unknown whether the identified inflammatory macrophage population has a unique function in MC migration after UV treatment.

4. The authors emphasize that the gender difference in melanocyte migration is due to differences in inflammatory responses, and they also found through single-cell sequencing data that UV radiation shifts macrophages in male mice towards an inflammatory phenotype. But since the authors didnt provide sufficient evidence to show that the inflammatory macrophages are uniquely functional in driving MC migration, the overall conclusion is not well support.

Reviewer #2 (Remarks to the Author):

In the manuscript of An et al the authors identified a methods platform using complementary techniques to promote epidermal melanocyte repopulation. They used C57BL/6 wildtype mice to understand the mechanisms of stimulation of McSC activation and migration induced by NBUBV. They found that the UVB-induced inflammation underlies McSC migration to the epidermis and that the gender-specific differences in inflammation levels support the differences in McSC translocation rates. They demonstrated that injecting intradermally PGE2, this promotes UVB induced inflammation and further addition of JAK inhibitor Ruxolitinib resulted in a significantly higher number of epidermal

melanocytes than any single therapy alone. They found by single-cell mRNA sequencing that prostaglandin signaling is responsible for the increased inflammatory response and McSC migration in males. They identified by single cell profiling the macrophages as likely mediators of McSC activation and migration due to UVB exposure.

General Comments:

The topic approached is interesting, the manuscript is well written and reads easily and the figures are high quality and very elaborated; the study uses a sophisticated testing platform and genetically engineered mice models, to track the migrating MCs stem cells and to overexpress genes. These are novel and crucial skin biology findings, with applicability to treat vitiligo, following a unique mechanistic concept, of stimulating directly the melanocyte stem cell reservoir by manipulating the inflammatory response.

Some comments appeared during my review, as summarized below.

Specific Comments:

---Can the authors comment if the increased inflammatory response observed in male mice would be expected in humans? Can these results generate the hypothesis that the repigmentation process should be greater in men vitiligo patients than in women?

---Page 5, Figure 3E: Pseudotime analysis of macrophage progression: the numeric and letters fonts for colors bar legend cannot be distinguished well.

---Are there different factors than hormonal differences in males vs females that can explain the increased McSC migration in male mice and the heightened skin inflammation?

---Page 9, line 2: Is there an explanation that male mice display profound keratinocyte differentiation as compared with female mice? Can be this the result of a difference in UV absorption by KCs in males vs females?

---Since the HF KCs are in a close anatomic and functional relationship with the MCs (like happens in epidermis), can authors comment how KCs are impacted by this NBUVB-induced inflammation? Aren't they key-facilitators of MCs migration?

---Page 20, line 20: I understand authors rationale of inducing a pro-inflammatory environment to stimulate the melanocyte stem cell migration in the HF. However, proposing for enhancing epidermal repigmentation, a combined treatment that includes an alternative reported to trigger genital but also extragenital vitiligo, can be approached with more caution- an examples is the paper "Imiquimod-induced hypopigmentation following treatment of periungual verruca vulgaris" with imiquimod inducing dorsal hands vitiligo-like depigmentation).

Responses to Reviewers

Reviewer #1 (Remarks to the Author):

The authors discovered that the migration rate of McSCs differs between male and female mice after UV treatment. Cox-2 and PGE2 were found to promote McSC proliferation and epidermal migration. The authors also identified a distinct subtype of macrophages that is involved in promoting MC migration. They proposed a working model that different inflammatory responses may explain the differences between males and females. The phenomenon is interesting, but the overall data quality needed to support their conclusions is not sufficient.

1. The author's results demonstrate increased MC migration and enhanced inflammatory response in male mice compared to female mice after UV radiation. Application of IMQ to both male and female mice resulted in increased MC migration. Therefore, the authors concluded there is a correlation between enhanced immune response and melanocyte migration. But UV treatment and IMQ treatment are two completely different things, using IMQ data to support immune cell difference is behind UV response difference between male and female is not logically coherent. To investigate whether immune cells are involved in UV induced MC migration difference between male and female mice the authors could try using immune-deficient mice, to see whether it affects melanocyte migration.

Thank you for your comments. We used IMQ in addition to UVB to boost the inflammatory response (Fig 2H and I), and found that IMQ promoted UVB-induced McSC migration. Combined with our prior manuscript (Moon et al. 2017) showing that the anti-inflammatory drug dexamethasone reduces McSC migration, we conclude that McSC migration is regulated by the inflammatory response.

As the reviewer notes, these data do not demonstrate an absolute dependence on the immune system for the sex difference in McSC migration. As suggested by reviewer, we utilized NSG immune-deficient mice to determine McSC migration differences in male and female mice under UVB irradiation. We found no statistical significance in the male versus female migration rate in NSG mice, and in most cases, no migration was found (Supplement Fig 2F). This data supports the strong regulatory influence immune cells have on the sex difference in McSC migration and in McSC migration in general. We believe that the residual myeloid population (Supplemental Figure 2E) in NSG mice may still have some capacity to facilitate migration in cases where migration does occur, which may be further examined in future studies.

2. The author's used a global Cre to knock out or overexpress Ptgs2, this cannot be used to draw conclusion of skin specific mechanisms. It is possible that the changes in melanocyte migration are caused by cells other than those residing in the skin. To support their conclusion, the authors need to use a skin cell-specific Cre mouse model or conduct skin graft experiments, for example the skin of mice with Ptgs2

knockout/overexpression can be transplanted onto wild-type mice, and ask whether the phenotype maintains.

Thank you for your comments. In this manuscript, we do not provide conclusions on skin or cell-specific mechanisms. Instead, we agree with the reviewer, in that cells that do not initially reside in the skin likely also contribute to this process. In this manuscript, we identified that the UVB-induced inflammatory microenvironment (including epidermal, immune, and fibroblast cells) has a role in promoting McSC migration to the epidermis. *Ptgs2* overexpression and global knockout mouse models were used to alter this inflammatory microenvironment and test for responses in McSC migration.

Ptgs2 is well-known to be expressed in multiple cell types, including both epithelial cells (Tripp et al. 2003) and components of the immune system, such as macrophages (Fig 4B). As suggested by the reviewer, to better understand the potential origins of the prostaglandins mediating this process, we have added data from mice overexpressing *Ptgs2* specifically in keratinocytes and myeloid cells (including macrophages and their progenitors, monocytes), through *Krt5-CreER; rosa26-*Isl-rtTA3* (KLP); Tre-Ptgs2*, and *Lysm-Cre; rosa26-*Isl-rtTA3*; Tre-Ptgs2 (MLP)* mice, respectively.

As shown now in Fig 5I and J, UVB-induced McSC migration demonstrated a statistically significant increase when *Ptgs2* is overexpressed in epidermal cells. *Ptgs2* overexpression in myeloid cells also showed a statistically significant increase in migration, albeit less pronounced than the effect observed in *KLP* (Fig 5K and L). However, this difference mirrors the relative intensity and cell number differences between the two Cre drivers, and thus likely reflects the relative amount of prostaglandins found in the inflammatory microenvironment (Supplement Fig 5H). These data do not necessarily indicate a defined cell-specific function at this point. Moreover, intradermal injection of dmPGE₂ led to an increase in McSC migration (Fig 7B), indicating that prostaglandin signaling can enhance McSC migration without necessarily relying on cell-type-specific origins. Regardless, these additional data show that an overall increased level of Cox-2 activity, upregulated Cox-2 expression in specific cells, or external dmPGE₂ administration is sufficient to increase McSC migration. This further supports that the difference in prostaglandin synthesis between male and female mice influences the differences in migration.

3. The author identified an inflammatory macrophage population in WT male mice after UV and in Cox-2 overexpression mice. However, in the case of Cox-2 overexpression mice, the author used clodronate liposomes to deplete all circulating macrophages without targeting this specific population of inflammatory macrophages, but the conclusions are drawn on the population of inflammatory macrophages. This conclusion need to be modified to reflect the experimental data. Without specific ablation experiment, it is unknown whether the identified inflammatory macrophage population has a unique function in MC migration after UV treatment.

We apologize for the lack of clarity on this point. We did not conclude that the pro-inflammatory macrophage population contributes to McSC migration. Rather, we identified a unique hybrid population found in UVB and Cox2 overexpression conditions. We have changed the section and figure titles to reflect this point. This title now emphasizes that a unique hybrid population was identified, but does not indicate a conclusion that this population is absolutely required for increased McSC migration. Unfortunately, we do not currently have a Cre-driver specific for this hybrid population.

However, we have added data showing that in animals that do not overexpress Ptgs2, macrophages are also important for McSC migration, similar to the overexpression model. In Figure 3J, we now demonstrate that the depletion of macrophages by Lym-Cre; iDtr results in decreased McSC migration.

As an additional note, macrophages are composed of very complex and dynamic cell sub-populations that are often unique to specific tissue types. Even though macrophages present pro-inflammatory and inflammatory resolving phenotypes in the skin, recent studies have determined that one macrophage state can rapidly transit to another. Given this propensity to quickly alter macrophage cell state, long-term ablation of a specific macrophage sub-population may be more challenging than it appears at first glance.

4. The authors emphasize that the gender difference in melanocyte migration is due to differences in inflammatory responses, and they also found through single-cell sequencing data that UV radiation shifts macrophages in male mice towards an inflammatory phenotype. But since the authors didn't provide sufficient evidence to show that the inflammatory macrophages are uniquely functional in driving MC migration, the overall conclusion is not well supported.

Thank you for your comment. We agree that the scRNA seq data is not sufficient to conclude that pro-inflammatory macrophages solely drive McSC migration. However, we are interpreting the results through a lens that considers the entire macrophage population as relative quantities of pro-inflammatory vs phagocytic, or pro-inflammatory vs phagocytic vs hybrid. This accounts for the ability of macrophages to alter phenotype, as discussed in the response to question 3.

Reviewer #2 (Remarks to the Author):

In the manuscript of An et al the authors identified a methods platform using complementary techniques to promote epidermal melanocyte repopulation. They used C57BL/6 wildtype mice to understand the mechanisms of stimulation of McSC activation and migration induced by NBUVB. They found that the UVB-induced inflammation underlies McSC migration to the epidermis and that the gender-specific differences in inflammation levels support the differences in McSC translocation rates. They

demonstrated that injecting intradermally PGE2, this promotes UVB induced inflammation and further addition of JAK inhibitor Ruxolitinib resulted in a significantly higher number of epidermal melanocytes than any single therapy alone. They found by single-cell mRNA sequencing that prostaglandin signaling is responsible for the increased inflammatory response and McSC migration in males. They identified by single cell profiling the macrophages as likely mediators of McSC activation and migration due to UVB exposure.

General Comments:

The topic approached is interesting, the manuscript is well written and reads easily and the figures are high quality and very elaborated; the study uses a sophisticated testing platform and genetically engineered mice models, to track the migrating MCs stem cells and to overexpress genes. These are novel and crucial skin biology findings, with applicability to treat vitiligo, following a unique mechanistic concept, of stimulating directly the melanocyte stem cell reservoir by manipulating the inflammatory response.

Some comments appeared during my review, as summarized below.

Specific Comments:

---Can the authors comment if the increased inflammatory response observed in male mice would be expected in humans? Can these results generate the hypothesis that the repigmentation process should be greater in men vitiligo patients than in women?

Thank you for your comments. Based on our data, we would expect a similar observation in humans, though no direct clinical evidence demonstrates a difference in the UVB response in patients, to our knowledge. However, there is indirect evidence suggesting men may have higher inflammatory responses than women when exposed to UVB irradiation. As described in our manuscript, in some cases, men have a higher melanoma incidence rate than women (Schwartz et al. 2019). This suggests higher McSC activation in men, which we believe can lead to melanoma (Moon et al, 2017). Additionally, inflammation is resolved faster in women than men in the case of cantharidin-mediated blistering (Rathod et al. 2023). More generally, studies have demonstrated that there are sex differences in inflammation and immune response between men and women in a variety of conditions (Klein and Flanagan 2016).

With regard to vitiligo, we have not found any reports using well-controlled sex-specific cohorts to compare repigmentation efficiency. Repigmentation relies on melanin production efficiency, the efficiency of transferring melanin to keratinocytes, and the longevity of keratinocytes retaining melanin. We did not assess these parameters between male and female mice. Nevertheless, this hypothesis would be worthwhile to explore in further clinical investigations and we have added this discussion in our revised manuscript (page 21, line 2-7)

---Page 5, Figure 3E: Pseudotime analysis of macrophage progression: the numeric and letters fonts for colors bar legend cannot be distinguished well.

Thank you for your recommendation. We have enlarged the fonts.

---Are there different factors than hormonal differences in males vs females that can explain the increased McSC migration in male mice and the heightened skin inflammation?

Thank you for your question. Yes, besides hormonal differences, the different level of DNA damage and reactive oxygen species (ROS) could contribute to the heightened skin inflammation in males. ROS and DNA damage are key signals that regulate tissue inflammation (Mittal et al. 2014; Arvanitaki et al. 2022). Studies have shown higher levels of single-strand breaks, alkali labile sites (Hofer et al. 2006), and ROS (Kim 2022) present in males. However, it is worth noting that both DNA damage and ROS level are regulated by the inflammatory response as well. Therefore, heightened skin inflammation shown in males could be the outcome of multifactorial interactions, and thus, cause vs effect is difficult to discern.

---Page 9, line 2: Is there an explanation that male mice display profound keratinocyte differentiation as compared with female mice? Can be this the result of a difference in UV absorption by KCs in males vs females?

Thank you for your questions. Following UVB irradiation, epidermal keratinocytes proliferate (epidermal hyperplasia) for a short period. The appearance of GO terms for keratinocyte differentiation reflects the onset of higher epidermal hyperplasia in males over females, which has been reported previously (Zhong et al. 2021). The hyperplasia thickness is related to the level of skin inflammation, which serves as another indicator of male vs female differences in inflammation levels.

The UV absorption coefficient is contingent upon the physical properties of the material. It is unlikely that there is a substantial difference in the molecular composition of male and female keratinocytes that could lead to distinct absorption coefficients.

---Since the HF KCs are in a close anatomic and functional relationship with the MCs (like happens in epidermis), can authors comment how KCs are impacted by this NBUVB-induced inflammation? Aren't they key-facilitators of MCs migration?

Thank you for your questions. Yes, keratinocytes are significantly affected by UVB and UVB-induced inflammation. UVB irradiation activates a keratinocyte DNA damage response and can cause keratinocytes to express cytokines, which recruit immune cells and modulate inflammation. In return, the recruited immune cells secrete growth factors, such as Igf-1 and Kgf-2 that can facilitate keratinocyte proliferation and epidermal hyperplasia.

Yes, keratinocytes have been found as the key facilitators of McSC migration. Under UVB irradiation, keratinocytes express Kitl and Wnt signaling to drive McSC migration.

However, the inflammatory role of UVB irradiation in vitiligo therapy has been under-appreciated, since vitiligo is an autoimmune disease, where immune suppression has been considered the key factor in developing a vitiligo therapy. One of the main points of this manuscript is to emphasize the importance of innate immune cells in driving McSC migration. In this manuscript, we determined that Cox-2 signaling and the inflammatory environment promote UVB-induced McSC migration. It is still unclear if this inflammatory environment targets McSCs *directly or indirectly by acting on keratinocytes*. As discussed in our manuscript (Page 22, line 7-11), future studies will tease apart the inflammatory impact arising from keratinocytes, melanocytes, and even fibroblasts to facilitate a better understanding and determine which cell type directly instructs this process.

---Page 20, line 20: I understand authors rationale of inducing a pro-inflammatory environment to stimulate the melanocyte stem cell migration in the HF. However, proposing for enhancing epidermal repigmentation, a combined treatment that includes an alternative reported to trigger genital but also extragenital vitiligo, can be approached with more caution- an examples is the paper "Imiquimod-induced hypopigmentation following treatment of periungual verruca vulgaris" with imiquimod inducing dorsal hands vitiligo-like depigmentation).

Thank you for the reference! Yes, an IMQ-based therapy would require extreme caution and so, we decided to remove this section of text. Instead, we discuss the impact of IMQ and potential mechanisms on vitiligo/vitiligo-like pathogenesis.

As the reviewer notes and the manuscript describes, IMQ has been used for some skin diseases, such as genital warts, and vitiligo-like depigmentation was frequently found in these patients. In our study, we found IMQ-induced inflammation dramatically facilitates UVB-induced McSC migration. Therefore, we hypothesized that IMQ-induced acute inflammation could activate McSCs. However, long-term IMQ application could cause McSC exhaustion and induce depigmentation. We hope this discussion will facilitate a better understanding of vitiligo-like pathogenesis due to IMQ treatment.

References

- Arvanitaki, E.S., Stratigi, K. and Garinis, G.A. 2022. DNA damage, inflammation and aging: Insights from mice. *Frontiers in Aging* 3, p. 973781.
- Hofer, T., Karlsson, H.L. and Möller, L. 2006. DNA oxidative damage and strand breaks in young healthy individuals: a gender difference and the role of life style factors. *Free Radical Research* 40(7), pp. 707–714.
- Kim, S.Y. 2022. Oxidative stress and gender disparity in cancer. *Free Radical Research*

56(1), pp. 90–105.

Klein, S.L. and Flanagan, K.L. 2016. Sex differences in immune responses. *Nature Reviews. Immunology* 16(10), pp. 626–638.

Mittal, M., Siddiqui, M.R., Tran, K., Reddy, S.P. and Malik, A.B. 2014. Reactive oxygen species in inflammation and tissue injury. *Antioxidants & Redox Signaling* 20(7), pp. 1126–1167.

Moon, H., Donahue, L.R., Choi, E., et al. 2017. Melanocyte stem cell activation and translocation initiate cutaneous melanoma in response to UV exposure. *Cell Stem Cell* 21(5), p. 665–678.e6.

Rathod, K.S., Kapil, V., Velmurugan, S., et al. 2023. Accelerated resolution of inflammation underlies sex differences in inflammatory responses in humans. *The Journal of Clinical Investigation*.

Schwartz, M.R., Luo, L. and Berwick, M. 2019. Sex differences in melanoma. *Current epidemiology reports* 6(2), pp. 112–118.

Tripp, C.S., Blomme, E.A.G., Chinn, K.S., Hardy, M.M., LaCelle, P. and Pentland, A.P. 2003. Epidermal COX-2 induction following ultraviolet irradiation: suggested mechanism for the role of COX-2 inhibition in photoprotection. *The Journal of Investigative Dermatology* 121(4), pp. 853–861.

Zhong, Q.-Y., Lin, B., Chen, Y.-T., et al. 2021. Gender differences in UV-induced skin inflammation, skin carcinogenesis and systemic damage. *Environmental toxicology and pharmacology* 81, p. 103512.

REVIEWERS' COMMENTS

Reviewer #1 (Remarks to the Author):

The authors have successfully addressed all of my questions.

Reviewer #2 (Remarks to the Author):

The authors addressed well all my comments, providing comprehensive answers. I do not have additional suggestions.